Manuscript prepared for Earth Syst. Dynam.
with version 2014/09/16 7.15 Copernicus papers of the LaTeX class copernicus.cls.
Date: 23 February 2018

# Midlatitude atmospheric circulation responses under 1.5°C and 2.0°C warming and implications for regional impacts

Camille Li[1,2], Clio Michel[1,2], Lise Seland Graff[3], Ingo Bethke[4,2], Giuseppe Zappa[5], Thomas J. Bracegirdle[6], Erich Fischer[7], Ben Harvey[5], Trond Iversen[3], Martin P. King[4,2], Harinarayan Krishnan[8], Ludwig Lierhammer[9], Daniel Mitchell[10], John Scinocca[11], Hideo Shiogama[12], Dáithí A. Stone[8,13], and Justin J. Wettstein[14,1,2]

[1]Geophysical Institute, University of Bergen, Bergen, Norway
[2]Bjerknes Centre for Climate Research, Bergen, Norway
[3]Norwegian Meteorological Institute, Oslo, Norway
[4]Uni Climate, Uni Research, Bergen, Norway
[5]Department of Meteorology, University of Reading, Reading, UK
[6]British Antarctic Survey, Cambridge, UK
[7]Institute for Atmospheric and Climate Science, ETH Zürich, Zürich, Switzerland
[8]Lawrence Berkeley National Laboratory, Berkeley, CA, USA
[9]German Climate Computing Center (DKRZ), Hamburg, Germany
[10]School of Geographical Sciences, University of Bristol, Bristol, UK
[11]Canadian Centre for Climate Modelling and Analysis, Environment and Climate Change Canada, Victoria, Canada
[12]National Institute for Environmental Studies, Tsukuba, Japan
[13]Global Climate Adaptation Partnership, Oxford, UK
[14]College of Earth, Ocean, and Atmospheric Sciences, Oregon State University, Corvallis, USA

*Correspondence to:* Camille Li (camille@uib.no)

**Abstract.**

This study investigates the global response of the midlatitude atmospheric circulation to 1.5°C and

2.0°C of warming using the HAPPI "Half a degree Additional warming, Prognosis and Projected Impacts" ensemble, with a focus on the winter season. Characterizing and understanding this response is critical for accurately assessing the near-term regional impacts of climate change and the benefits of limiting warming to 1.5°C above pre-industrial levels, as advocated by the Paris Agreement of the United Nations Framework Convention on Climate Change (UNFCCC). The HAPPI experimental

design allows an assessment of uncertainty in the circulation response due to model dependence and internal variability. Internal variability is found to dominate the multi-model mean response of the jet streams, storm tracks and stationary waves across most of the midlatitudes; larger signals in these features are mostly consistent with those seen in more strongly forced warming scenarios. Signals that emerge in the 1.5°C experiment are a weakening of storm activity over North America, an inland

shift of the North American stationary ridge, an equatorward shift of the North Pacific jet exit, and an equatorward intensification of the South Pacific jet. Signals that emerge under an additional 0.5°C of

warming include a poleward shift of the North Atlantic jet exit, an eastward extension of the North Atlantic storm track, and an intensification on the flanks of the Southern Hemisphere storm track. Case studies explore the implications of these circulation responses for precipitation impacts in the Mediterranean, western Europe and on the North American west coast, paying particular attention to possible outcomes at the tails of the response distributions. For example, the projected weakening of the Mediterranean storm track emerges in the 2°C warmer world, with exceptionally dry decades becoming five times more likely.

## 1 Introduction

There is growing urgency to understand the near-term impacts of climate change for low-end warming targets. This need arises from the Paris Agreement's aim to "strengthen the global response to the threat of climate change by keeping a global temperature rise this century well below 2 degrees Celsius above pre-industrial levels and to pursue efforts to limit the temperature increase even further to 1.5 degrees Celsius" (UNFCCC, 2015). In order to better assess the associated impacts, the scientific community devised the "Half a degree Additional warming, Prognosis and Projected Impacts" (HAPPI) initiative (Mitchell et al., 2017; http://www.happimip.org/).

The purpose of this study is to examine the midlatitude atmospheric circulation response in the HAPPI experiments, which represent worlds that are 1.5°C and 2.0°C warmer than pre-industrial conditions (or 0.7°C and 1.2°C warmer than present conditions; see section 2). With these relatively weak warming scenarios, nonlinear or threshold responses such as those that may arise from changes in atmospheric circulation are particularly important in the context of the Paris Agreement. The HAPPI experiments were designed to quantify the potential benefits of a mitigation effort to reduce warming by an extra half a degree, i.e., the climate impacts avoided by limiting warming to 1.5°C compared to 2.0°C (Mitchell et al., 2017). For the purposes of this study, two aspects of the HAPPI experiments bear mention. First, the use of large ensembles allows an investigation of the spread in the responses, in particular, changes in the tails and shapes of their distributions. This is especially important when the responses of interest are related to the midlatitude atmospheric circulation, which features large internal variability (Deser et al., 2012; Shepherd, 2014). Second, the use of common sea surface temperature (SST) and sea ice conditions allows us to isolate the consensus atmospheric response across models.

Despite the expectation that the mean response of the midlatitude atmospheric circulation will be weak in these experiments, the present study nevertheless furnishes critical baseline knowledge for the HAPPI project. Specifically, it documents the large-scale background climate changes being used to compute indices for extreme events and as input for impact models (e.g., Baker, submitted 2017; Jacob et al., 2018; Mitchell, submitted 2017; Shiogama, submitted 2017; Wehner et al., in press

2018). In this way, it provides information that can help inform, interpret and corroborate results of other, more impacts-focused, HAPPI studies.

Section 2 briefly describes the philosophy of the HAPPI experiments, the experimental setup, and model output, as well as the statistical framework used to analyze the HAPPI ensemble. Section 3
presents the multi-model mean circulation responses under 1.5°C of warming and the additional 0.5°C of warming, with a focus on the wintertime, the season during which midlatitude dynamics – and the associated poleward energy transport – are most vigorous. Section 4 presents case studies to illustrate possible implications of the model results for regional climate change by looking beyond the multi-model mean responses to the spread in the responses (in essence, by exploring precipita-
tion changes associated with outcomes at the tails of the response distributions). Finally, section 5 raises some discussion points and concluding remarks. In addition, a supplement contains figures showing the model biases compared to ERA-Interim reanalysis and GPCP data (Dee et al., 2011; Adler et al., 2003; S1), the multi-model mean responses for the summer season (S2), and the 2.0°C response relative to present climate (S3).

**2 Methods and data**

**2.1 Experimental setup**

The HAPPI modelling experiments allow the investigation of atmospheric responses under weak warming scenarios, their associated uncertainties, and the resulting impacts. The experiments follow a protocol similar to current (and proposed) climate experiments, particularly those within the
70 International CLIVAR Climate of the 20th Century Plus Detection and Attribution (C20C+ D&A) project (Folland et al., 2014; Stone et al., in prep 2018), to best exploit synergies. In contrast to scenarios based on emissions or greenhouse gas concentration trajectories used by the Coupled Model Intercomparison Project (CMIP), the HAPPI approach is based on a global temperature constraint, using large ensembles to allow for comparison of extremes and to feed into impact models (see Fig.
1 in Mitchell et al., 2017).

Mitchell et al. (2017) document the setup of the Tier 1 HAPPI experiments, performed using the atmosphere-only models listed in Table 1. The three Tier 1 experiments simulate a present decade (PD, 2006–2015), a decade that is 1.5°C warmer than pre-industrial conditions (approximately 0.7°C warmer than PD), and a decade that is 2.0°C warmer than pre-industrial conditions (approximately
80 1.2°C warmer than PD).

Ensemble members for all models within each experiment are 10-year long simulations differing only in their initial conditions. They are forced with time-varying SST, sea ice and anthropogenic radiative forcings (due to greenhouse gases, aerosols, land use, land cover) estimated for the period of interest (PD, 1.5°C or 2.0°C). Natural radiative forcings are prescribed according to observed PD
values in all three experiments. The future experiment forcings come from the lower end CMIP5

scenarios, RCP2.6 and RCP4.5, which produce a global mean temperature response at the end of the century (2091–2100) of approximately 1.5°C and 2.5°C, respectively, above pre-industrial levels (Fig. 2 in Mitchell et al., 2017; Collins et al., 2013). All models include active land components such that surface temperature over land points is not fixed.

The main setup points for the three experiments are listed below; additional details on how the forcings were constructed can be found in Mitchell et al. (2017). Note that, for each experiment, all models use identical ocean boundary conditions derived from the CMIP5 multi-model mean.

  – Present decade (PD) experiment: observed SST, sea ice, atmospheric greenhouse gas concentrations, aerosols, ozone, land use and land cover for present conditions (2006–2015) are used.
The SSTs and sea ice include interannual variability.

  – 1.5°C experiment: SST changes, calculated as the difference between a 1.5°C warmer world (CMIP5 multi-model mean for RCP2.6 scenario over the period 2091–2100) and the present decade (CMIP5 multi-model mean for RCP8.5 scenario over the period 2006–2015), are added to the PD SSTs. Sea ice concentrations are adjusted for consistency with the warmer SSTs.
Atmospheric greenhouse gas concentrations, aerosols, ozone, land use and land cover are set to 2095 values from RCP2.6.

  – 2.0°C experiment: SST and sea ice cover are calculated in a similar way as those for the 1.5°C experiment, except that changes are determined using a weighted sum of the RCP2.6 and RCP4.5 scenarios over the period 2091–2100 to correspond to a 2.0°C warmer world.
Atmospheric $CO_2$ concentrations are set to a weighted average of values from RCP2.6 and RCP4.5 in the 2.0°C experiment; other atmospheric greenhouse gases, aerosols, ozone, land use and land cover are set to 2095 values from RCP2.6.

Monthly variables used in the study are surface air temperature (K), precipitation (mm d$^{-1}$), zonal wind at 850 hPa ($u850$, m s$^{-1}$), zonal wind at 250 hPa ($u250$, m s$^{-1}$), meridional wind at 250 hPa
($v250$, m s$^{-1}$), geopotential height at 500 hPa ($Z500$, m) and temperature at 850 hPa and 200 hPa ($T850$ and $T200$, K). Stationary waves are calculated as departures from the zonal mean ($Z500*$, $v250*$). Daily variables are filtered with a 2–6 day bandpass filter to isolate synoptic-scale variability in mean sea level pressure (MSLP′, hPa), zonal wind at 250 hPa ($u250'$, m s$^{-1}$) and meridional wind at 250 hPa ($v250'$, m s$^{-1}$). These are used to calculate storm track metrics: the low-level storm tracks
are defined as the standard deviation of MSLP′, and the upper-level storm tracks are defined via eddy kinetic energy (EKE = $u250'^2 + v250'^2$, m$^2$ s$^{-2}$ ). Model biases of the PD experiment compared to ERA-Interim and GPCP are shown in the supplement (S1).

The temperature responses are consistent with those from the RCP scenario simulations performed under CMIP5 (see Fig. 12.12 Collins et al., 2013). Maps of surface air temperature responses (Fig. 1)
look reasonable relative to the RCP scenarios (Collins et al., 2013), featuring the expected amplification of warming over land and in the Arctic. There is model consensus everywhere on the *sign* of

**Table 1.** Models used in study, including the number of ensemble members run for each of the PD, 1.5°C and 2.0°C experiments.

| Model | Horizontal resolution | Vertical levels | Members / experiment | References |
|-------|----------------------|-----------------|---------------------|------------|
| CAM4 | $1.9° \times 2.5°$ | 26 | 501 | Neale et al. (2013) |
| CanAM4 | T63 (96 × 192) | 35 | 100 | von Salzen et al. (2013) |
| MIROC5 | T85 (128 × 256) | 40 | 100 | Shiogama et al. (2013) |
| ECHAM6.3-LR | T63 (96 × 192) | 47 | 100 | Lierhammer et al. (2017) |
| NorESM1-Happi | $0.94° \times 1.25°$ | 26 | 125 | Bentsen et al. (2013) |
| | | | | Iversen et al. (2013) |
| | | | | Kirkevåg et al. (2013) |

the forced response, which is to be expected since the imposed SST and sea ice boundary conditions are identical across all models for each experiment. The black dots indicate regions where there is substantial model dependence in the *magnitude* of the forced response (see section 2.2 for details
on this metric). In these regions (e.g., near the sea ice edge), the response exhibits very small internal variability, and model dependence is hence proportionally large. "Magnitude agreement" by this measure improves over high latitude regions with the additional 0.5°C of warming, indicated by a reduction in dots in 2.0°C−PD (Fig. S3.1a) and even 2.0°C−1.5°C (Fig. 1b) compared to 1.5°C−PD (Fig. 1a). The troposphere generally warms due to increased greenhouse gas concentrations, with
stronger near-surface warming in the Arctic and stronger upper tropospheric warming in the tropics, and the stratosphere cools (not shown).

## 2.2   Statistical framework

The HAPPI multi-model ensemble is analaysed and interpreted using a two-way analysis of variance (ANOVA) framework (see Sansom et al., 2013, and appendix). Each model's climate change
response is computed as the difference between the future (1.5°C or 2.0°C) and present (PD) mean climate, which is estimated by averaging across all the available ensemble members for each model (see Table 1). The multi-model mean response ($\beta$) is then obtained as the average of the individual models' future responses.

The signal-to-noise ratio in the climate change response ($\beta/\sigma$) is evaluated relative to the noise
due to internal decadal climate variability ($\sigma$ is the standard deviation across the 10-year members). Following Sansom et al. (2013), one value of $\sigma$ is obtained for the whole multi-model ensemble by pooling together variations in decadal mean climate across all the ensemble members. A signal-to-noise ratio $\beta/\sigma$ greater than 1 implies that the amplitude of the climate change response is larger than decadal climate variability, suggesting a substantial climatological impact in the region. A $\beta/\sigma$

less than 1 implies that internal variability dominates, but there could still be a non-negligible impact depending on the region.

Finally, we quantify the robustness of the response across models. There are many methods of assessing model robustness, differing in their assumptions about the ensemble's statistical properties, the way they estimate internal variability, and their definitions of "robust" (see Box 12.1 in Collins et al., 2013). Two measures providing complementary information are used here. The first is a standard measure of *sign* consensus, defined here as four out of five models agreeing on the sign of the forced response. Note that we do not generally expect a lack of sign consensus in the HAPPI data set because the forced response is well sampled (there is a large ensemble for each individual model). The second measure provides information on how well models agree on the *magnitude* of the forced response. This is estimated via a metric, $f^2$, representing a ratio of variance explained by "structural" uncertainty (model dependence) over the variance explained by internal variability. The variance estimates are derived from the ANOVA framework, and not simply the spread across models or across members (see appendix for details). $f^2 < 1$ implies that internal climate variability is the dominant source of uncertainty in the multi-model projections. If model dependence is proportionally small ($f^2 < 1$) where there is no sign consensus, this indicates an agreement that there is a weak (or no) response; if model dependence is proportionally large ($f^2 > 1$) where there is sign consensus, this indicates that the magnitude of the response is still quite uncertain. Note that, because the boundary conditions do not account for decadal SST variations, both $f^2$ and the signal-to-noise ratio will tend to be inflated in the HAPPI experiments compared to observations or coupled model experiments, particularly in the tropics.

The multi-model mean precipitation results for 1.5°C of warming (1.5°C−PD) and an additional 0.5°C of warming (2.0°C−1.5°C) are shown in Figure 2 for winter, and in Figure 3 for summer. The top panels show the multi-model mean response $\beta$, with the hatching masking out regions where there is no model consensus (fewer than four out of five models agree) on the sign of the response. In shading in the bottom panels is the signal-to-noise ratio $\beta/\sigma$ (the sign corresponds to the sign of the response). The patterns of $\beta/\sigma$ and $\beta$ are similar, but relative magnitudes between locations can differ if the spread associated with decadal variability is very different. Model agreement on the sign of the response is generally good, even where the response $\beta$ is weak, with only a few exceptions (black dots in the Canadian Archipelago, Kara Sea, Caspian Sea). Note that a location with high signal-to-noise ratio may exhibit model dependence (black dots) if one or two models dominate the response (e.g., 1.5°C−PD response in Arctic regions shown in Fig. 2c).

## 3 Midlatitude circulation features: multi-model mean response

This section presents an overview of the multi-model mean circulation response in the 1.5°C experiment (1.5°C−PD) and with an additional 0.5°C of warming (2.0°C−1.5°C). Note that the 1.5°C

experiment represents 1.5°C of warming relative to pre-industrial climate, but only 0.7°C of warming compared to the PD. While the focus of this study is the winter season (Northern Hemisphere DJF, Southern Hemisphere JJA), features of interest in other seasons will be discussed as appropriate. In the Northern Hemisphere, winter is the season with strongest surface baroclinicity, associated with fast jet streams, high-amplitude stationary waves and maximum-intensity storm tracks, all of

which contribute to the large poleward transport of energy (Chang et al., 2002; Trenberth and Stepaniak, 2003). In the Southern Hemisphere, we include discussion of other seasons since stratospheric ozone depletion is an important driver of summer changes (Thompson et al., 2011), and baroclinicity and storm tracks are strong all year round (van Loon, 1967; Hoskins and Hodges, 2005). The supplement contains the summer versions of all figures in this section (S2) and the 2.0°C−PD responses

(S3).

Generally, circulation changes are quite weak, but somewhat consistent with those observed in the more strongly forced, coupled climate change simulations from CMIP3 and CMIP5 (e.g., Ulbrich et al., 2008; Chang et al., 2012; Simpson et al., 2014; Woollings et al., 2012; Eichler et al., 2013; Barnes and Polvani, 2013). As expected, internal variability is responsible for most of the uncertainty

in the responses shown here (Deser et al., 2012; Shepherd, 2014). We first describe the responses of the jet streams, storm tracks and stationary waves, and then draw comparisons with previous studies.

**Jet streams.** The midlatitude jet stream responses are small and mostly barotropic (observed in $u850$ Fig. 4 and $u250$ Fig. 5), with some differences appearing in the 2.0°C experiment that are

not apparent in the 1.5°C response. The main signal is an equatorward shift of the North Pacific jet exit, and a slight poleward shift of the North Pacific jet entrance, consistent with the jet changes in RCP8.5 (Simpson et al., 2014). These changes add roughly linearly with the warming from PD to 1.5°C, and from 1.5°C to 2.0°C (compare with Fig. S3.3 and S3.4). In contrast, the North Atlantic jet exhibits different response patterns in 1.5°C−PD and 2.0°C−1.5°C. The 1.5°C experiment shows

very little change relative to PD, although the jet entrance shifts slightly equatorward. In the 2.0°C experiment, the jet exit shows a poleward shift and slight extension, associated with a strengthening of westerlies over Europe and a weakening of westerlies over the Mediterranean (Fig. 4). In the Southern Hemisphere, very little response is noted in the 1.5°C experiment other than an extension and slight equatorward shift of the South Pacific jet at upper levels (Fig. 5a). The 2.0°C−1.5°C

response is a strengthening across the South Pacific and a slight poleward shift at upper levels that brings the jet back to its PD position (Fig. S3.3, S3.4).

These experiments do not show the jets clearly migrating or expanding poleward, as found in the more strongly forced CMIP3 (Delcambre et al., 2013) and CMIP5 (Woollings and Blackburn, 2012; Barnes and Polvani, 2013) warming scenarios. The zonal-mean jet shifts are close to zero

in all seasons and ocean sectors (Fig. 6). The most robust signals, though still weak, are (based on 95% confidence intervals from 10,000 bootstrapped realizations of the full 926-member HAPPI

ensemble): a poleward shift of the North Atlantic jet other than in winter; an equatorward shift of the North Pacific jet during the cold season, particularly in the 1.5°C experiment; an equatorward shift of the Southern Hemisphere jet in austral summer in the 1.5°C experiment; and a poleward shift of the Southern Hemisphere jet in the shoulder seasons in the 2.0°C experiment. In the North Atlantic and Southern Hemisphere, the poleward shift during the shoulder seasons is consistent with the results of Barnes and Polvani (2013). The North Pacific jet exhibits especially large spread (box-and-whiskers in Fig. 6 indicate the spread across all members for all models), partly related to the longitudinally varying nature of the response (Fig. 4, 5). The winter zonal-mean signal is dominated by an equatorward shift of the jet exit, partly offset by a poleward shift of the jet entrance (see Fig. 7 for 2.0°C−PD); the summer zonal-mean signal averages over large and opposing shifts in the jet entrance and exit regions. Still, the lack of dots over the North Pacific in Fig. 4c,d indicates that the uncertainty in the forced response is small compared to variations in the 10-year means.

**Storm tracks.** The storm track responses are also small, but broadly consistent with previous results from CMIP3 and CMIP5 (e.g., Ulbrich et al., 2008; Chang et al., 2012; Zappa et al., 2013). In the Northern Hemisphere, lower-level storm activity weakens overall, but the change is generally small except over North America and at very high latitudes (Harvey et al., 2012; Chang, 2013) (Fig. 8). Most of the weakening occurs from PD to 1.5°C; the additional half a degree to 2.0°C does little other than to intensify slightly the tail end of the North Atlantic storm track (compare Fig. 8 and S3.6), consistent with previous studies (e.g., Harvey et al., 2012). In the Southern Hemisphere, lower-level storm activity shows weak, patchy changes in the 1.5°C experiment, and still weak but uniform increases in the 2.0°C experiment (see Chang et al., 2012). The increased storm activity is concentrated on the flanks of the storm track (most apparent in Fig. S3.6), suggesting possible influences on the Antarctic coast to the poleward side, and parts of South America, South Africa and southern Australia to the equatorward side. Upper-level storm activity shows a poleward shift and upward expansion of the storm tracks (e.g., Yin, 2005) in both hemispheres, with the response getting larger from 1.5°C to 2.0°C (Fig. 9 and Fig. S3.7). The fact that the upper-level responses exhibit similar patterns for 1.5°C−PD and 2.0°C−1.5°C while the lower level responses are quite different supports previous studies suggesting that upper tropospheric eddy activity is less sensitive to local surface forcing (Harvey et al., 2015; Ciasto et al., 2016).

**Stationary waves.** The midlatitude stationary wave response is relatively consistent from the 1.5°C experiment to the 2.0°C experiment (Fig. 10, S3.5). The main feature is over the west coast of North America, where the stationary ridge shifts inland, weakening in the southwest and strengthening in the northeast. This ridge response enhances southerlies off the west coast and northerlies in the interior of the continent. The response shows good agreement with results from previous studies, which propose changes in the tropics or jet strength as possible mechanisms (Selten et al., 2004;

Haarsma and Selten, 2012; Simpson et al., 2016). In the 2.0°C experiment, the ridge response is
slightly stronger, while some new (rather small) features emerge at high northern and southern lati-
tudes.

Many of the circulation responses listed above have similar patterns in 2.0°C and 1.5°C, but in
some cases, the response patterns change from the initial 0.7°C of warming (1.5°C−PD) to the ad-
ditional 0.5°C of warming (2.0°C−1.5°C). In the Northern Hemisphere, one reason may be that
changes in the low-level equator-to-pole temperature contrasts are more important in shaping the
1.5°C response than the 2.0°C response. Greenhouse warming sharpens the upper-level tempera-
ture gradient and pushes the jet/storm track poleward, while near-surface Arctic amplification of
this warming weakens the low-level temperature gradient and pushes the jet/storm track equator-
ward (e.g., Bengtsson and Hodges, 2006; Brayshaw et al., 2008; Butler et al., 2010; Lu et al., 2010;
Graff and LaCasce, 2012; Harvey et al., 2014; Deser et al., 2015; Shaw et al., 2016), with various
possibilities for the precise mechanism (Lorenz and DeWeaver, 2007; Chen et al., 2008; Kidston
et al., 2011; Michel and Rivière, 2014). The idea that the low-level gradient $\Delta T850$ has greater
leverage in the 1.5°C experiment is supported by the fact that this gradient weakens more with the
initial 0.7°C of warming than with the additional 0.5°C of warming (Fig. 11c for ECHAM6.3-LR,
Table 2 for multi-model means). Maanwhile, the upper-level gradient $\Delta T200$ strengthens more with
the additional 0.5°C (Fig. 11a, Table 2) except in NorESM1-Happi (Table 3). As the world warms,
the strengthening upper-level gradients eventually "win": in stronger warming scenarios, the zonal-
mean jets in the North Atlantic, North Pacific and Southern Hemisphere shift poleward (Barnes
and Polvani, 2013), although the exit region of the North Pacific jet actually shifts equatorward in
winter to weaken the signal in that sector and season (Simpson et al., 2014). Even in the HAPPI
experiments, the changes from 1.5°C to 2.0°C are in the poleward direction in all sectors (for the
North Atlantic, the jet shift is actually "more poleward"; for other sectors, it is "less equatorward";
see Fig. 4, 5, 6). Harvey et al. (2014) found that, in the coupled RCP scenarios, low-level gradi-
ent changes are tied to sea-ice changes, and upper-level gradients changes are tied to tropical SST
changes (via the tropical lapse rate). Because sea ice and SST are prescribed in the HAPPI ensemble,
there is generally good agreement in the response of these gradients across all models (see Table 3
for the Northern Hemisphere). The exception is the 1.5°C−PD change in $\Delta T200_{NH}$, possibly due
to model-dependence of the tropical lapse rate response.

Another reason for different response patterns in 1.5°C−PD and 2.0°C−1.5°C is simply due to
the setup of the experiments. In the 2.0°C experiment, atmospheric $CO_2$ concentrations are higher
and SSTs are warmer than in the 1.5°C experiment. Aerosols, ozone and land use, however, are set
to the same values in the two warming experiments (taken from the year 2095 in the RCP2.6 sce-
nario). Thus, the influence of $CO_2$ and SSTs changes relative to the influence of other anthropogenic
forcings going from 1.5°C to 2.0°C. Ozone recovery in particular warms the Antarctic stratosphere

during austral summer, opposing the tendency of greenhouse warming to sharpen the upper-level gradient $\Delta T200_{\text{SH}}$ (Gerber and Son, 2014). The ozone effect dominates the Southern Hemisphere $1.5°C-PD$ response, but is absent in the $2.0°C-1.5°C$ response, thus accounting for the strong cancellation between $1.5°C-PD$ and $2.0°C-1.5°C$ responses in all the summertime circulation features (see supplement S2).

**Table 2.** Multi-model mean response of equator-to-pole temperature contrast (°C) in winter at 850 hPa and 200 hPa. The 5th percentile, median, and 95th percentile responses are shown. The percentile values and medians are first computed for each model, then a member-weighted multi-model mean is taken. Positive values indicate a stronger contrast between the warm tropical atmosphere and cold polar atmosphere with warming. See the caption of Figure 11 for definitions of the temperature contrasts.

|  | $\Delta T850_{\text{NH}}$ | $\Delta T200_{\text{NH}}$ | $\Delta T850_{\text{SH}}$ | $\Delta T200_{\text{SH}}$ |
|---|---|---|---|---|
| $1.5°C-PD$ |  |  |  |  |
| 5% | -1.166 | -0.280 | -0.443 | 0.398 |
| median | -0.719 | 1.015 | -0.111 | 1.001 |
| 95% | -0.239 | 2.399 | 0.220 | 1.616 |
| $2.0°C-1.5°C$ |  |  |  |  |
| 5% | -0.700 | -0.107 | -0.301 | 0.408 |
| median | -0.262 | 1.209 | 0.045 | 0.965 |
| 95% | 0.190 | 2.444 | 0.380 | 1.509 |

**Table 3.** Median responses from individual models of the equator-to-pole temperature contrast (°C) in winter at 850 hPa and 200 hPa for the Northern Hemisphere only. Positive values indicate a stronger contrast between the warm tropical atmosphere and cold polar atmosphere with warming. See the caption of Figure 11 for definitions of the temperature contrasts.

| Model | $\Delta T850_{\text{NH}}$ | | $\Delta T200_{\text{NH}}$ | |
|---|---|---|---|---|
|  | $1.5°C-PD$ | $2.0°C-1.5°C$ | $1.5°C-PD$ | $2.0°C-1.5°C$ |
| CAM4 | -0.749 | -0.245 | 1.162 | 1.258 |
| CanAM4 | -0.726 | -0.421 | 0.476 | 1.150 |
| MIROC5 | -0.577 | -0.233 | 1.036 | 1.175 |
| ECHAM6.3-LR | -0.658 | -0.260 | 0.457 | 1.051 |
| NorESM1-Happi | -0.755 | -0.228 | 1.289 | 1.210 |

## 4  Case studies

Here, we present case studies to investigate the idea that the documented circulation changes (Fig. 4–10), however weak, may have regional impacts through precipitation (Fig. 2). The cases were selected based on the fact that they show associated changes in circulation and precipitation, yet illustrate that the link between dynamics and impacts under global warming can be different in different locations.

### 4.1  Mediterranean

The Mediterranean region is thought to be especially vulnerable to drought risk under climate change (Giorgi 2006, Diffenbaugh and Giorgi 2012). Characterized by hot, dry summers, it depends critically on precipitation during the winter season. Under greenhouse warming produced by the RCP scenarios, the region is projected to undergo drying due to a weakening of the Mediterranean storm track (Collins et al., 2013; Zappa et al., 2015a), as measured by reductions in both cyclone activity (Bengtsson et al. 2006; Lionello and Giorgi 2007; Raible et al. 2010) and low-level westerly winds (Zappa et al., 2015b). An additional drying effect stems from reduced moisture convergence into the region as a result of changes in the mean flow, in particular the development of an anomalous near-surface high that is linked to subsidence and mass flux divergence (Seager et al., 2014a).

In the HAPPI experiments, Mediterranean drying appears to set in somewhere between 1.5°C and 2.0°C of warming according to the multi-model mean (Fig. 12c,d). The wintertime drying is a dynamical effect of climate change: a weakening of the Mediterranean trough, which suppresses storm activity, is reflected in a weakening of the mean westerlies over North Africa ($u850$; Fig. 12b) and an anomalous high in the sea level pressure field over the Mediterranean (SLP; Fig. 15d). The relevant circulation changes, and hence the dynamical effect, are extremely weak in the 1.5°C experiment (Fig. 12a and 15c), thus we do not expect precipitation to change much (Fig. 12c). Interestingly, the northern part of the region actually becomes slightly wetter in 1.5°C as part of a general precipitation increase across western Europe (see section 4.2). This is consistent with the idea that, without the dynamical effect, the main influence of climate change would be to increase the amount of precipitation carried by each cyclone, making the region wetter (Zappa et al., 2015a).

Whilst the mean drying of the Mediterranean in the 2.0°C experiment is quite clear (5–10% reductions in precipitation across the region in Fig. 12d), internal variability still plays an important role. Zappa et al. (2015b) previously found a strong, linear relationship between seasonal anomalies in $u850$ and seasonal anomalies in precipitation – a relationship that holds on interannual time scales in observations and historical simulations, as well as on climate change time scales. This $u850$-precipitation relationship also holds across all the models used in this study for both experiments (Fig. 12e,f). From the scatterplots, it is clear that the responses exhibit considerable spread sampling both positive and negative precipitation changes. In the 1.5°C experiment (Fig. 12e), the distribu-

tion straddles zero, making the multi-model mean nearly zero, while in the 2.0°C experiment, the distribution shifts towards more drying (i.e., towards the lower left quadrant of Fig. 12f).

Given the large internal variability, it is perhaps more instructive to examine the distribution of responses rather than the median or mean response. Compositing over members that exhibit the
strongest 5% of wind responses in the 2.0°C experiment ("strong" is defined as most weakening of the wintertime westerlies over the box in Fig. 13a,b, identified for each model first then composited over all models ~~such that a "strong" response is defined in the sense of the changes in the multi-model mean in Fig. 12d~~), we see 0.37 mm d$^{-1}$ less precipitation over the region compared to the PD, which represents a reduction of 27% in winter precipitation compared to the PD clima-
tology (Fig. 13a). Exceptionally dry decades corresponding to the 5th precipitation percentile in the 2.0°C experiment occur <1% of the time in the PD experiment on average, with considerable model spread (Table 4); for example, there is no change for CanAM4 (the 5% limit in the 2.0°C distribution maps onto 5% in the PD distribution), while for CAM4 and MIROC5, dry decades become drier than any in the PD distribution (the 5% limit in the 2.0°C distribution maps onto 0% in the
PD distribution). On the other end of the spectrum, the members with the weakest wind responses actually show slightly more precipitation over the region (0.21 mm d$^{-1}$; Fig. 13b). However, there is overall a decreased probability for wet winters, shown by the fact that precipitation rates for the 5% wettest decades in the 2.0°C experiment represent a larger share in the PD distribution for all models (Table 4). Histograms of the changing rainfall distribution for sample locations (Fig. 14a) also show
a consistent shift towards smaller precipitation rates across most of the region (Rabat, Morocco and Athens, Greece), despite very weak changes in the mean/median precipitation values (Table 5). Lisbon, Portugal, which is situated in the northern part of the region, exhibits a shift towards slightly wetter winters in the 1.5°C experiment before the drying sets in at 2.0°C.

**Table 4.** Mapping of extreme decades in winter (DJF) precipitation from the 2.0°C experiment to the PD experiment. Values are the PD percentages corresponding to the 5% driest (5th percentile) and 5% wettest (95th percentile) limits in the 2.0°C experiment. Zero indicates that the 2.0°C extreme precipitation values do not exist in the PD distribution. The last column is a member-weighted multi-model mean. The Mediterranean region is defined by the box in Fig. 12d; the Euro-Atlantic region is defined by the box in Fig. 15b.

|  | CAM4 | CanAM4 | ECHAM6.3-LR | MIROC5 | NorESM-Happi | mean |
|---|---|---|---|---|---|---|
| Mediterranean |  |  |  |  |  |  |
| 5% driest | 0 | 5.0 | 1.0 | 0 | 0.8 | 0.8 |
| 5% wettest | 16.6 | 11.0 | 23.0 | 8.0 | 5.6 | 15.4 |
| Euro-Atlantic |  |  |  |  |  |  |
| 5% driest | 50.5 | 66.0 | 29.0 | 31.0 | 51.2 | 47.8 |
| 5% wettest | 0 | 0 | 0 | 0 | 0 | 0 |

**Table 5.** Decadal mean winter (DJF) precipitation (mm d$^{-1}$) for sample locations from the CAM4 ensemble corresponding to the histograms in Fig. 14a.

|  |  | Experiment |  |  |
| --- | --- | --- | --- | --- |
|  | PD | 1.5°C | 2.0°C |  |
| Athens, | 2.102 | 2.069 | 2.000 | mean |
| Greece | 2.105 | 2.065 | 1.997 | median |
|  | 1.775 | 1.718 | 1.651 | 5% |
|  | 2.433 | 2.437 | 2.395 | 95% |
| Lisbon, | 1.475 | 1.501 | 1.342 | mean |
| Portugal | 1.461 | 1.497 | 1.333 | median |
|  | 1.106 | 1.122 | 0.979 | 5% |
|  | 1.918 | 1.955 | 1.751 | 95% |
| Rabat, | 0.545 | 0.521 | 0.421 | mean |
| Morocco | 0.536 | 0.511 | 0.412 | median |
|  | 0.355 | 0.334 | 0.254 | 5% |
|  | 0.743 | 0.750 | 0.603 | 95% |

The signal beginning to emerge in a 2.0°C world is effectively a poleward expansion of climate zones accompanying a poleward expansion of the Hadley cell (Lu et al., 2007). The Köppen-Geiger scheme for climate zones (Fig. 14b) classifies the Mediterranean today as a temperate region (Csa, Csb; characterized by dry summers), containing some arid steppe microclimates (BSk, BSh). Under global warming, the North African arid belt pushes northwards, and the Mediterranean temperate zone spreads into Europe. Places that may be most susceptible to winter rainfall deficits are those situated where the drying is most pronounced (e.g., see Iberian Peninsula in Fig. 13a; Lisbon histograms in Fig. 14a, Table 5) or near the transition to the Sahara Desert (e.g., Rabat histograms in Fig. 14a, Table 5).

## 4.2 Euro-Atlantic region

The atmospheric circulation response to climate change in the Euro-Atlantic sector exhibits substantial uncertainty due to large internal variability and the influence of Arctic amplification of global warming, which is pronounced in the region. In CMIP scenarios with midrange to strong warming, previous studies have noted an eastward extension of the winter storm track into Europe, with higher cyclone frequencies in central Europe and reduced cyclone frequencies in Scandinavia and the Mediterranean (Chang et al., 2012; Harvey et al., 2012; Zappa et al., 2013). These storm track

changes are associated with a robust poleward shift of the North Atlantic jet, primarily in the exit region, in all seasons except winter (Woollings and Blackburn, 2012; Barnes and Polvani, 2013).

In the HAPPI experiments, there are signs that similar changes are emerging, although in some cases not until 2.0°C warming. The North Atlantic jet exit shifts poleward in the 2.0°C experiment (Fig. 4, 5). In the 1.5°C experiment, near-surface Arctic amplification has considerable leverage (as noted in the discussion surrounding Fig. 11), resulting in a slight equatorward shift of the jet entrance. At low levels, the multi-model mean exhibits weaker westerlies over North Africa in both warming experiments, with stronger westerlies over most of the European continent in the 2.0°C experiment (Fig. 12a,b). Finally, the storm-track exit is slightly enhanced at 1.5°C and extends eastward into Europe at 2.0°C, with considerable spread due to internal variability (Fig. 15a,b).

These circulation changes are tied to precipitation impacts over Europe, but in contrast to the Mediterranean, they reinforce (rather than oppose) the thermodynamic effect. Western Europe has reliable, year-round precipitation and sees increasing precipitation with warming, evident in both the 1.5°C and 2.0°C experiments (Fig. 2). This is partly due to the thermodynamic increase in tropospheric moisture content, which alters moisture fluxes and leads to wet regions getting wetter (Held and Soden, 2006). On top of this, the dynamic effect (extension of the storm track in 2.0°C; Fig. 15b) also contributes to wetting the region. The signal-to-noise ratio of the storm track response is small, reflecting the tendency of models to place the climatological storm track in slightly different locations (Fig. S1.5). Despite this, the resulting influence on precipitation is clear. Mean or median precipitation in the region under the storm track extension (indicated by the box in Fig. 15a,b) increases by 4.0% to 4.5% per degree of warming in both the 2.0°C−1.5°C and 1.5°C−PD multi-model mean responses (not shown). More striking is the change in the tails of the precipitation distribution. Exceptionally dry decades corresponding to the 5th precipitation percentile in the 2.0°C experiment sit near the median (48th percentile) of the PD distribution, while exceptionally wet decades corresponding to the 95th precipitation percentile in the 2.0°C experiment are characterized by precipitation rates completely outside the PD distribution (Table 4).

Finally, the MSLP responses are also small, but consistent with the jet changes. The 1.5°C experiment shows a negative anomaly over the Atlantic (Fig. 15c) where we note an equatorward shift of the jet entrance. There is a slight increase near Iceland and a slight decrease over the Azores, suggesting lower values of the North Atlantic Oscillation (NAO) index. The 2.0°C experiment shows a dipole over the European sector (negative anomalies over Scandinavia, positive over the Mediterranean; Fig. 15d) consistent with a poleward shift of the jet exit. However, the response over the NAO centres of action is very weak (see stippling). We calculate a daily wintertime (DJF) NAO index for each ensemble member by subtracting the standardized SLP at Reykjavik from the standardized SLP at the Azores (the standardization uses the multi-model PD ensemble as the reference time series). The daily NAO index is averaged over each member to produce a distribution of decadal NAO variability for each model (Fig. 15e). The changes in the NAO are model-dependent, but a robust feature

is a shift towards more negative NAO values in the 1.5°C experiment (equatorward shift of jet entrance) and a slight shift back towards more positive NAO values in 2.0°C−1.5°C (poleward shift of the jet exit).

## 4.3 North American west coast

The North American west coast is the last case study, chosen to illustrate the different effects that circulation changes may have on precipitation at different locations. In the strong warming RCP8.5 scenario, models project a drying of the interior southwest of the continent and a wetting to the north (including the west coast from California up to Alaska) during the winter half-year due primarily to changes in moisture convergence associated with the mean flow (Seager et al., 2014b). Consistent with the analysis of Seager et al. (2014b), the mean circulation response in the HAPPI warming experiments is also responsible for altering humidity flux in the region. Both the 1.5°C and 2.0°C experiments show a deepening and eastward extension of the Aleutian low (Fig. 16a,c), along with specific humidity fluxes at 850 hPa that intensify towards North America and deflect northward relative to the PD (Fig. 16b,d).

The altered Aleutian low produces two notable features in the multi-model mean circulation response: (1) an extension and equatorward shift of the North Pacific jet exit that enhances westerlies just offshore of the west coast (seen in the $u250$ field in Fig. 5), and (2) an inland migration of the North American ridge that enhances southerlies into Alaska (Fig 10; see also Selten et al. (2004); Haarsma and Selten (2012); Simpson et al. (2016)). We examine the relationship between these wind changes and precipitation at two locations along the west coast. Because 2.0°C of warming produces changes that are similar to, but larger than, those with 1.5°C of warming, only the 2.0°C results are shown here.

On the central west coast (southern pink box in Fig 17), the jet ($u250$) and stationary ridge ($v250*$) changes appear to have competing (or at least offsetting) effects on precipitation. In members with the strongest $u250$ response (most positive westerly anomaly offshore in black box in Fig. 17a), the central west coast becomes wetter and there is a slight enhancement of low-level storm activity just offshore (Fig. S3.6a,c). In members with the strongest $v250*$ response (inland ridge shift producing southerly anomaly in black box in Fig. 17c), there is little change (Fig. 17c). Precipitation signals associated with jet changes are larger than those associated with ridge changes. Within the spread produced by internal variability, the jet response can be positive or negative, such that the members with the weakest $u250$ responses yield drying along the central west coast (Fig. 17b). Regional winter precipitation changes by 13% of the climatological PD value between members with the strongest and weakest jet responses.

On the southern coast of Alaska (northern pink box in Fig 17), both the jet and stationary ridge changes are associated with increased precipitation. However, it is a strong $v250*$ response that produces the most wetting (Fig. 17c), consistent with the relationship between stronger offshore

southerlies and a wetting of northwestern North America (Seager et al., 2014b). Internal variability, while still present, is not important to the overall sign of the precipitation response: members at the other end of the spectrum still become wetter in the region, although only very slightly (Fig.17d). Regional precipitation changes by 4% of the climatological PD value between members with the strongest and weakest ridge responses.

## 5  Discussion and concluding remarks

This study presents an overview of the global midlatitude circulation response to 1.5°C and 2.0°C of warming compared to pre-industrial conditions using the HAPPI ensemble. The main findings are as follows:

- The large ensembles reveal complex responses to the HAPPI forcings, with internal (decadal) variability playing an important role in the ensemble spread.

- In the 1.5°C experiment, there is a weakening of storm activity over North America, an inland shift of the North American stationary ridge, an equatorward shift of the North Pacific jet exit, and an equatorward intensification of the South Pacific jet. With an additional 0.5°C of warming, most response features are enhanced but some new ones emerge, most notably a poleward shift of the North Atlantic jet exit, an eastward extension of the North Atlantic storm track into Europe, and an intensification on the flanks of the Southern Hemisphere storm track.

- Mediterranean: The projected drying of the region emerges in the 2.0°C world, along with a weakening of the Mediterranean storm track. The HAPPI ensemble simulates 5-10% local reductions in average winter precipitation over the region, with exceptionally dry decades (1st percentile or a 1-in-100 event in the PD experiment) becoming five times more likely (5th percentile or a 1-in-20 event) in the 2.0°C experiment.

- Western Europe: The region becomes increasingly wet from 1.5°C to 2.0°C, although the main circulation changes (slight poleward shift in the North Atlantic jet and eastward extension of the low-level storm track) only appear in the 2.0°C experiment. The HAPPI ensemble suggests that average winter precipitation today (48th percentile in the PD experiment) will represent very dry decades in the future (5th percentile or a 1-in-20 event in the 2.0°C experiment), and future wet extremes will fall outside the PD distribution.

- North American west coast: This region gets wetter with warming. Precipitation increases over the central west coast are mainly associated with enhanced westerlies at the North Pacific jet exit, while precipitation increases over the southern Alaskan coast are mainly associated with an inland shift of the North American stationary ridge.

While the unique experimental design of the HAPPI ensemble is an advantage, it also introduces
limitations that must be considered when interpreting and applying our results. The specified SST
and sea ice conditions allow for large ensembles, but there is high uncertainty on future SST and sea
ice changes. At least by constructing the forcings as an ensemble mean over all CMIP5 models, po-
tential errors in these fields are likely smaller than errors from an individual model. But local features
of the forcing seem to heavily influence the regional changes in these low-end warming scenarios,
as evidenced by the complex, nonlinear behaviour in the $1.5°C-PD$ and $2.0°C-1.5°C$ responses.
Furthermore, the single realization of SST and sea ice patterns as well as the lack of atmosphere-
ocean-ice coupling offer a restricted view of ocean-driven variability and internal variability, which
is sure to influence the simulated climate variability. Comparisons with related coupled experiments
(e.g., Sanderson et al., 2017; Iversen et al., submitted 2017) would be a useful exercise for evaluating
whether coupling changes the results presented here.

There remain many interesting large-scale questions to explore using the HAPPI ensemble, in-
cluding deeper investigations into how individual model biases affect the responses and their spread;
contrasting drivers of forced changes in jet latitude and jet speed (McGraw and Barnes, 2016; Baker
et al., 2017; Bracegirdle et al., 2018); particular regions where the additional half a degree of warm-
ing yields impacts not seen in the $1.5°C$ experiment; and potential consequences of uncertainties in
future SST and sea ice changes.

## Appendix A: Statistical framework

Following Sansom et al. (2013), the $f^2$ metric is evaluated as the ratio between the variance in the
multi-model ensemble due to differences in the model responses and the variance due to internal
climate variability. These variances are determined by fitting two different ANOVA frameworks to
the output of the HAPPI simulations.

The first framework is the two-way ANOVA framework, which accounts for model dependence
(structural uncertainty) in both the present decade (PD) mean state (through the term $\alpha_m$) and in
the climate change response (through the term $\gamma_m$). In particular, if $y_{msr}$ is the decadal mean state
simulated by model $m$ for scenario $s$ and ensemble member $r$, it is defined in the two-way framework
as

$$y_{msr} = \underbrace{\mu + \alpha_m}_{\substack{\text{PD climate} \\ \text{in model } m}} + \underbrace{\beta_s + \gamma_{ms}}_{\substack{\text{climate change} \\ \text{response} \\ \text{in model } m}} + \epsilon_{msr}, \tag{A1}$$

where $\mu$ is the expected mean state in the PD climate; $\alpha_m$ is the difference between the mean PD cli-
mate simulated by model $m$ and the expected PD climate $\mu$; $\beta_s$ is the expected mean climate change
response; $\gamma_{ms}$ is the difference between the mean response in scenario $s$ simulated by model $m$ and
the the expected climate change response $\beta_s$; and $\epsilon_{msr}$ represents internal variability, which is as-
sumed to be normally distributed with constant variance, i.e., $\epsilon_{msr} \overset{iid}{\sim} N\left(0, \sigma^2\right)$. The scenario index

$s$ can refer to the present decade (PD) or future (labelled F; in our case, either 1.5°C or 2.0°C) experiments, and the framework is subject to the constraint $\beta_{\mathrm{PD}} = \gamma_{m\mathrm{PD}} = 0$ for all models. The two-way framework captures all the variance in the multi-model ensemble due to model dependence, apart from the internal variability assigned to $\epsilon_{msr}$. We define $R_\gamma^2$ as the coefficient of determination of the two-way framework – the proportion of the total variability explained by the two-way framework. $R_\gamma^2$ accounts for model dependence in both the PD climate and the climate change response, so that $(1 - R_\gamma^2)$ is the remaining variance, which is due to internal climate variability.

To isolate the model dependence of the climate change response, we need to introduce a second framework. The additive ANOVA framework is a simplified version of the two-way ANOVA framework that assumes there is *no model dependence* in the climate change response, i.e., $\gamma_{m\mathrm{F}} = 0$ for all models. Equation A1 then becomes

$$y_{msr} = \mu + \alpha_m + \beta_s + \epsilon_{msr}. \tag{A2}$$

We define $R_\alpha^2$ as the coefficient of determination of the additive framework, which is related to model dependence in the PD mean state only ($\alpha_m$). The difference $R_\gamma^2 - R_\alpha^2$ is then the proportion of variance due to differences in the model responses.

Combining information from the two-way and additive frameworks, the ratio of variance explained by structural uncertainty (model dependence) in the climate response to that explained by internal climate variability may thus be quantified as

$$f^2 = \frac{(R_\gamma^2 - R_\alpha^2)}{(1 - R_\gamma^2)} \tag{A3}$$

*Acknowledgements.* CM and LSG contributed equally to the study. This work was supported by the Norwegian Research Council projects no. 261821 HappiEVA (IB, LSG, TI, CM), no. 231716 jetSTREAM (CM), no. 255027 DynAMiTe (MPK); the U.S. Department of Energy, Office of Science, Office of Biological and Environmental Research contract DE-AC02-05CH11231 (HK, DAS); funding from the Bundesministerium für Bildung und Forschung BMBF (LL). We acknowledge the Norwegian Metacenter for Computational Science and Storage Infrastructure (NOTUR and Norstore Projects NN2345k and NS2345k); the European Centre for Medium-Range Weather Forecasts for providing the ERA-Interim data; and the NOAA/OAR/ESRL PSD (Boulder, Colorado, USA) for providing the GPCP precipitation data at www.esrl.noaa.gov/psd. Finally, we thank M. Esch, K.-H. Wieners, S. Hagemann, T. Mauritsen from MPI-M for technical support with ECHAM6.3-LR; S. Legutke, E. Madonna, S. Sobolowski for helpful discussions; and two anonymous reviewers for their constructive suggestions.

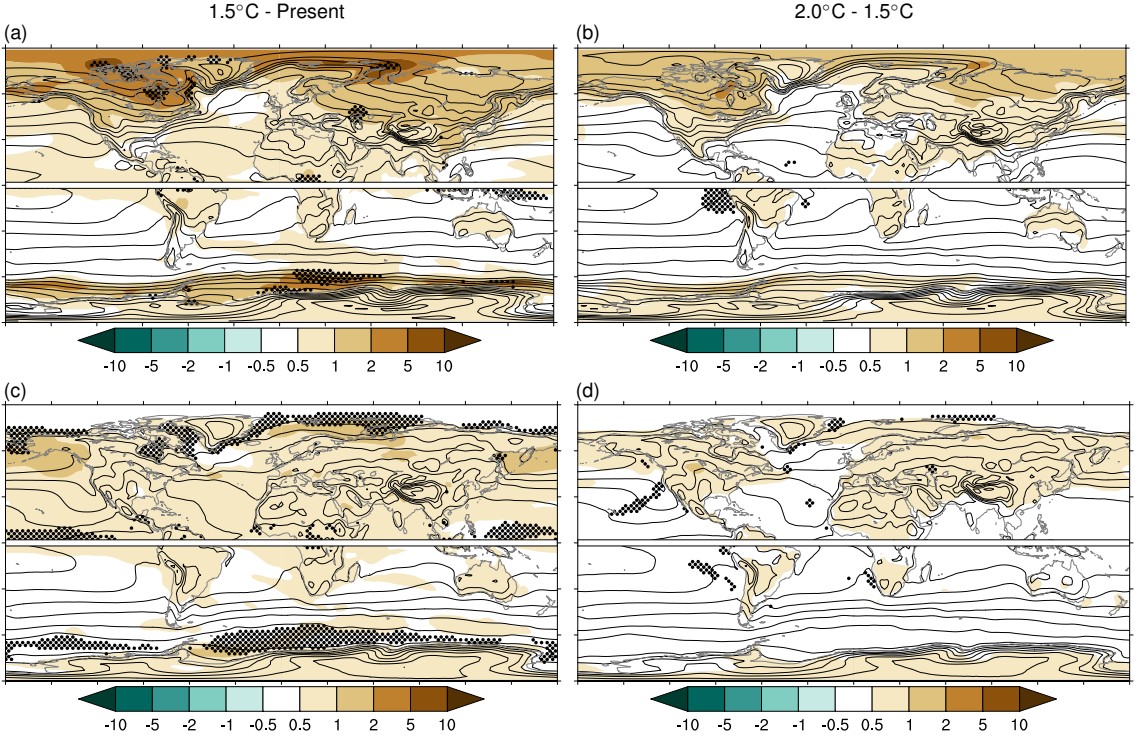

**Figure 1.** Multi-model mean response of surface air temperature for 1.5°C−PD (left) and 2.0°C−1.5°C (right). Top panels show winter (Northern Hemisphere DJF, Southern Hemisphere JJA) responses (shading; units K) along with the climatology (contour interval 5 K) for the (a) PD and (b) 1.5°C experiments. Bottom panels show the same for summer (Northern Hemisphere JJA, Southern Hemisphere DJF) along with the climatology (contour interval 5 K) for the (c) PD and (d) 1.5°C experiments.. There is model consensus on the sign of the response everywhere. Black dots mask out regions where there is model dependence, i.e., the models do not agree on the magnitude of the response ($f^2 > 1$; see section 2.2 for details).

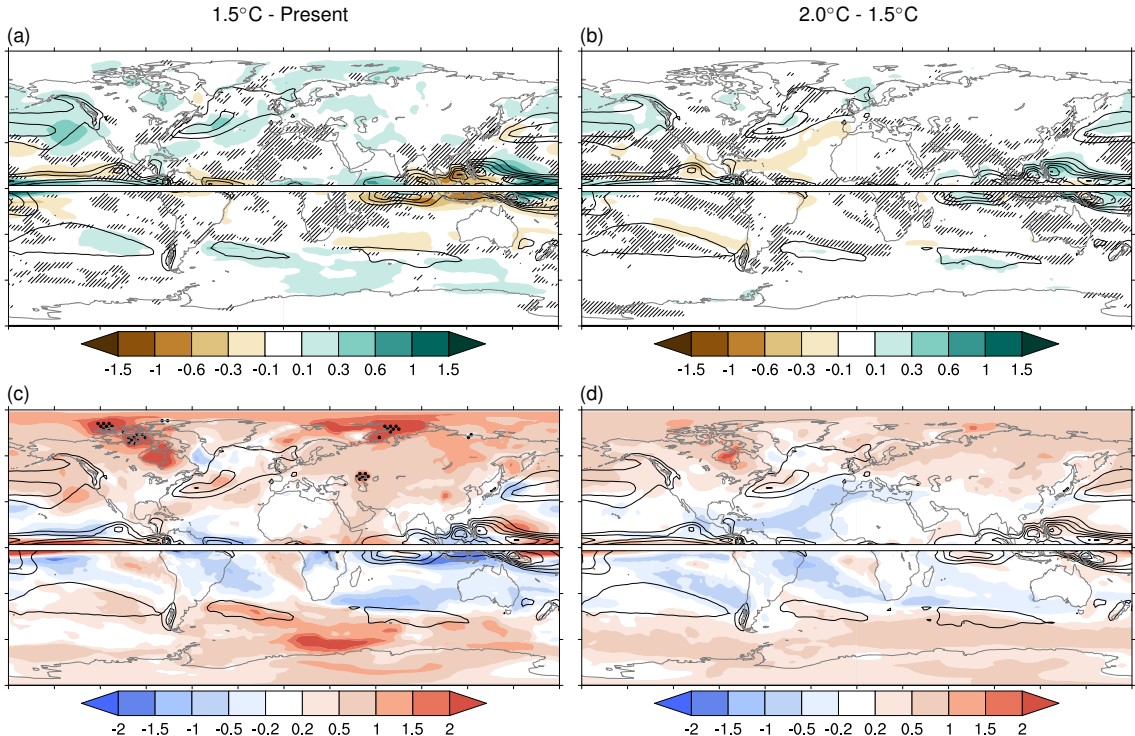

**Figure 2.** Multi-model mean response of winter (Northern Hemisphere DJF, Southern Hemisphere JJA) precipitation for 1.5°C−PD (left) and 2.0°C−1.5°C (right). Top panels show responses (shading; units mm d$^{-1}$) along with the climatology (contour interval 2 mm d$^{-1}$ starting from 4 mm d$^{-1}$) for the (a) PD and (b) 1.5°C experiments. Bottom panels show signal-to-noise ratio $\beta/\sigma$, where the sign corresponds to the sign of the response, along with the climatology (contour interval 2 mm d$^{-1}$ starting from 4 mm d$^{-1}$) for the (c) 1.5°C and (d) 2.0°C experiments. In (a) and (b), hatching masks out regions where there is no model consensus (fewer than four out of five models agree) on the sign of the response. In (c) and (d), black dots mask out regions where the models do not agree on the magnitude of the response ($f^2 > 1$).

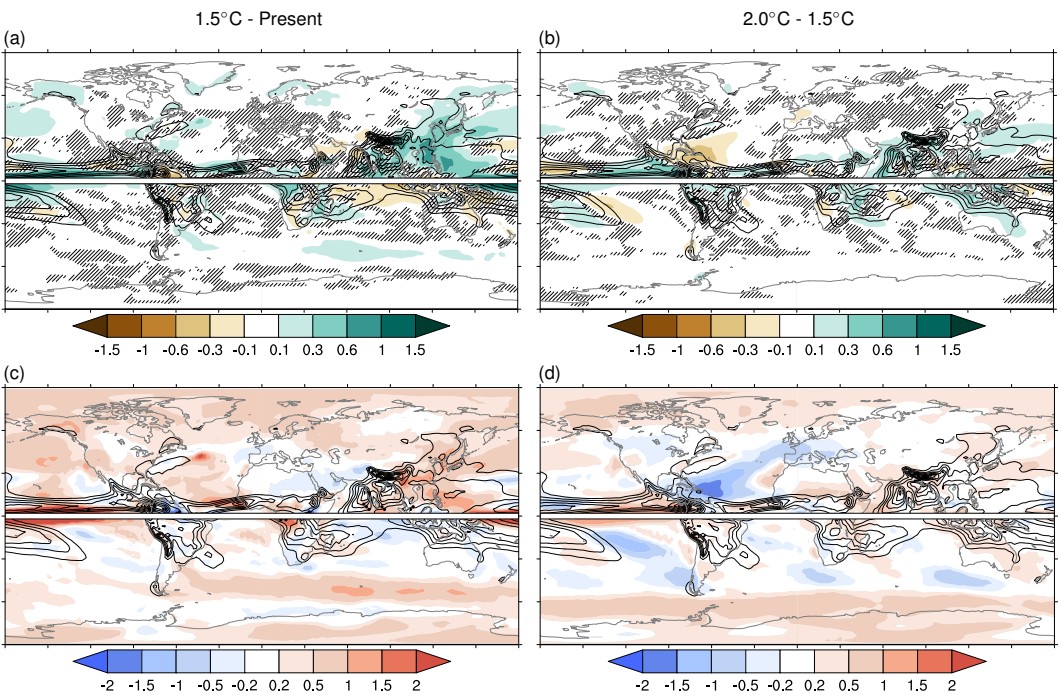

**Figure 3.** As in Figure 3 but for summer (Northern Hemisphere JJA, Southern Hemisphere DJF) precipitation.

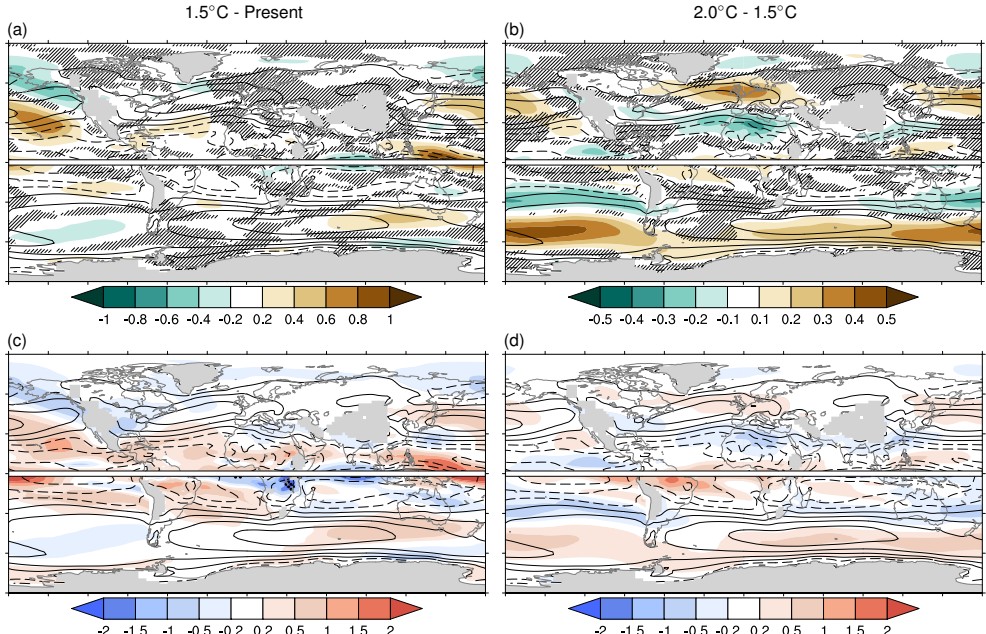

**Figure 4.** Multi-model mean response of winter (Northern Hemisphere DJF, Southern Hemisphere JJA) zonal wind at 850 hPa ($u850$) for 1.5°C−PD (left) and 2.0°C−1.5°C (right). Top panels show responses (shading; units m s$^{-1}$) along with the climatology (contour interval 4 m s$^{-1}$) for the (a) PD and (b) 1.5°C experiments. Note the different colour scale for (a) and (b). Bottom panels show signal-to-noise ratio $\beta/\sigma$, where the sign corresponds to the sign of the response, along with the climatology (contour interval 4 m s$^{-1}$) for the (c) 1.5°C and (d) 2.0°C experiments. In (a) and (b), hatching masks out regions where there is no model consensus (fewer than four out of five models agree) on the sign of the response. In (c) and (d), black dots mask out regions where the models do not agree on the magnitude of the response ($f^2 > 1$). Grey shading indicates regions of high topography intersecting the plotted variable.

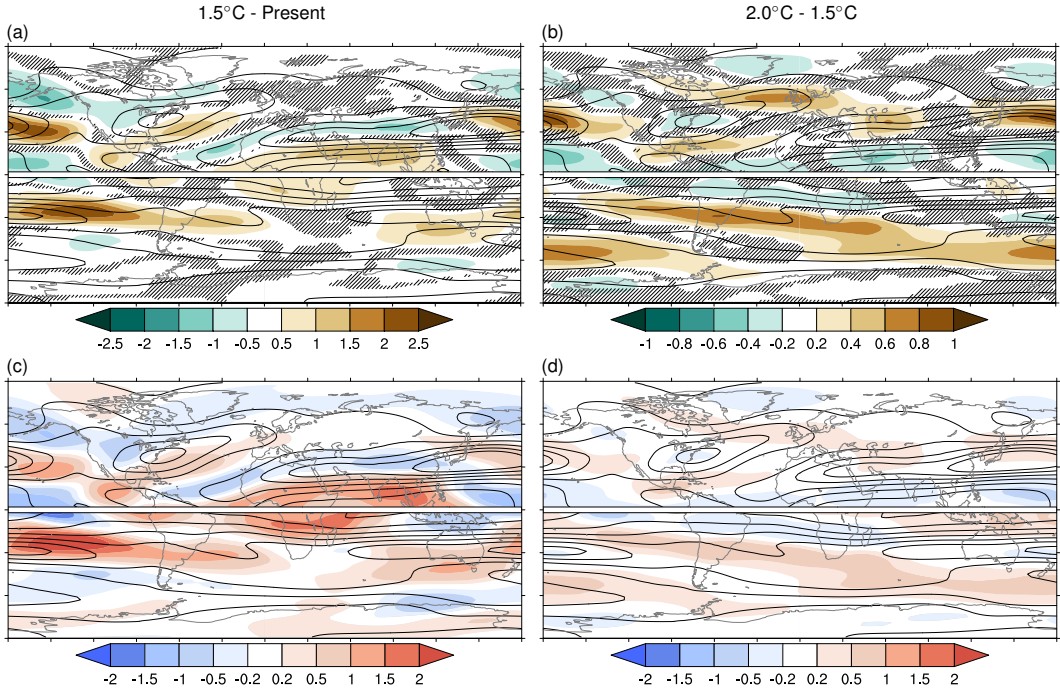

**Figure 5.** Multi-model mean response of winter (Northern Hemisphere DJF, Southern Hemisphere JJA) zonal wind at 250 hPa ($u250$) for $1.5°C-PD$ (left) and $2.0°C-1.5°C$ (right). Top panels show responses (shading; units m s$^{-1}$) along with the climatology (contour interval 10 m s$^{-1}$) for the (a) PD and (b) 1.5°C experiments. Note the different colour scale for (a) and (b). Bottom panels show signal-to-noise ratio $\beta/\sigma$, where the sign corresponds to the sign of the response, along with the climatology (contour interval 10 m s$^{-1}$) for the (c) 1.5°C and (d) 2.0°Cc experiments. In (a) and (b), hatching masks out regions where there is no model consensus (fewer than four out of five models agree) on the sign of the response. In (c) and (d), black dots mask out regions where the models do not agree on the magnitude of the response ($f^2 > 1$).

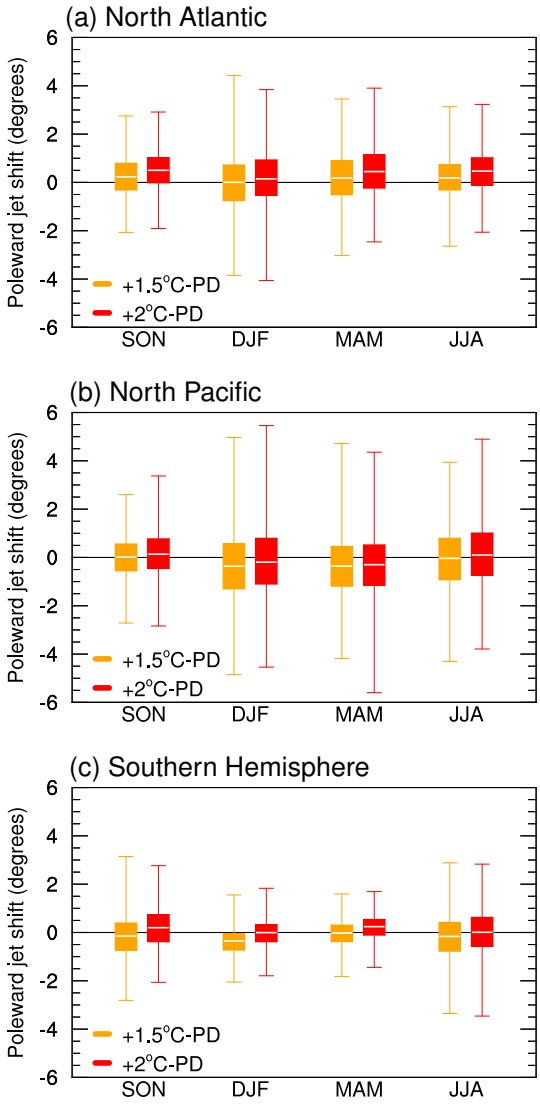

**Figure 6.** Multi-model mean shift of the eddy-driven jet in the 1.5°C (orange) and 2.0°C (red) experiments relative to the PD experiment for each season in the North Atlantic sector (60°W–0), North Pacific sector (135°E–235°E) and Southern Hemisphere. The white horizontal line in the boxes indicates the median, the boxes indicate the interquartile range, and the whiskers indicate the spread across all ensemble members for all models. Jet latitude is determined according to the method of Woollings et al. (2010). The mean shifts are very close to, but different from, zero based on 95% confidence intervals from 10,000 bootstrapped realizations of the full HAPPI ensemble (926 members over all models), with the exception of the following seasons/sectors: for 1.5°C−PD, North Atlantic DJF, North Pacific JJA/SON, Southern Hemisphere MAM; for 2.0°C−PD, North Pacific DJF/JJA/SON, Southern Hemisphere DJF/JJA.

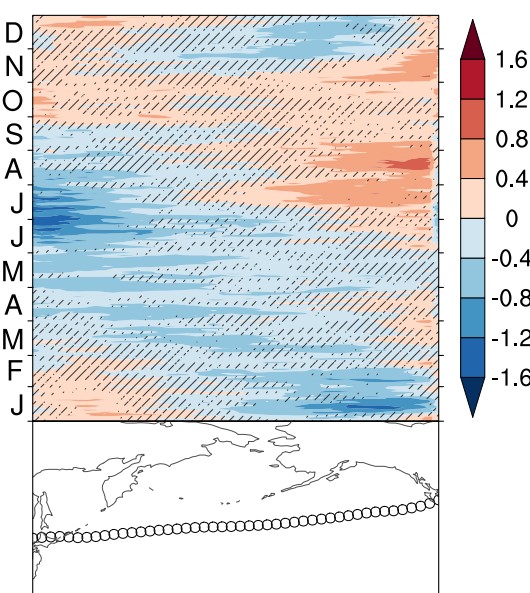

**Figure 7.** Multi-model mean shift of the ~~winter~~ North Pacific eddy-driven jet (shading, in degrees latitude) in the 2.0°C experiment compared to the PD experiment as a function of longitude and time. Hatching masks out regions where there is no model consensus (fewer than four out of five models agree) on the sign of the response. The bottom panel shows the climatological position of the jet in the PD experiment. Jet latitude is calculated following Simpson et al. (2014).

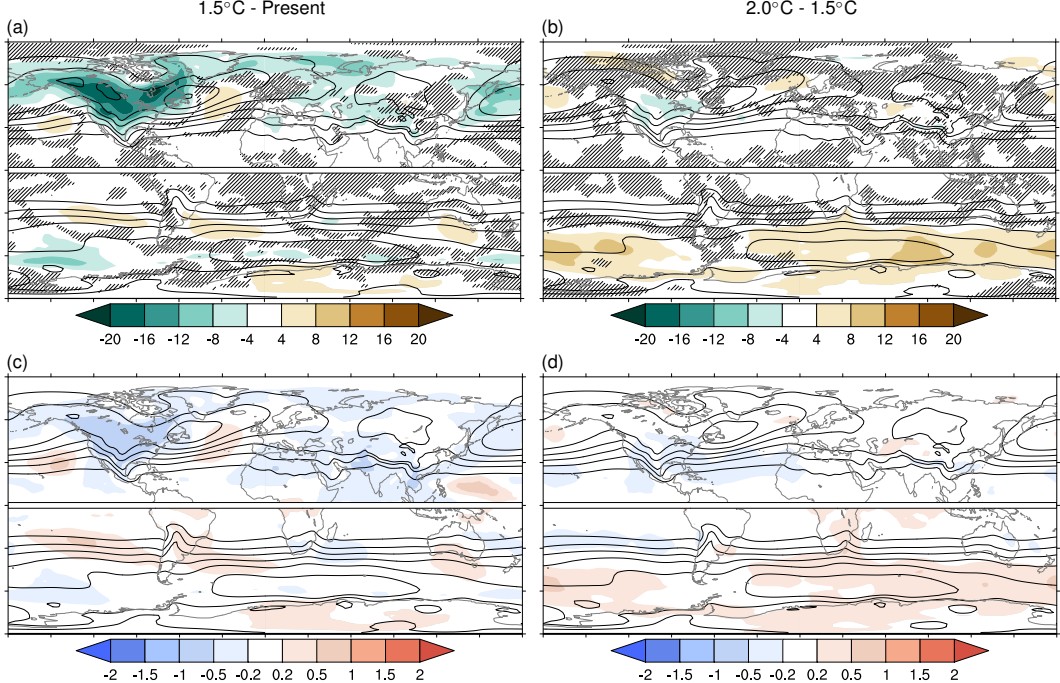

**Figure 8.** Multi-model mean response of winter (Northern Hemisphere DJF, Southern Hemisphere JJA) low-level MSLP storm tracks for 1.5°C−PD (left) and 2.0°C−1.5°C (right). Top panels show responses (shading; units hPa) along with the climatology (contour interval 100 hPa) for the (a) PD and (b) 1.5°C experiments. Bottom panels show signal-to-noise ratio $\beta/\sigma$, where the sign corresponds to the sign of the response, along with the climatology (contour interval 100 hPa) for the (c) 1.5°C and (d) 2.0°C experiments. In (a) and (b), hatching masks out regions where there is no model consensus (fewer than four out of five models agree) on the sign of the response. In (c) and (d), black dots mask out regions where the models do not agree on the magnitude of the response ($f^2 > 1$). The storm tracks are defined as the standard deviation of bandpass filtered daily MSLP.

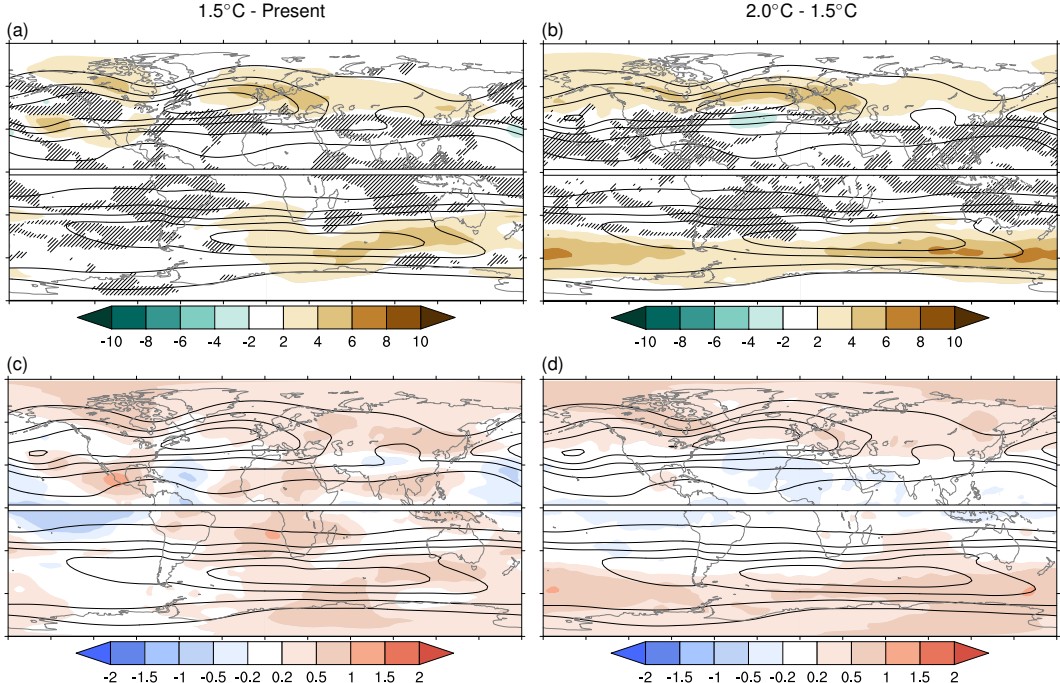

**Figure 9.** Multi-model mean response of winter (Northern Hemisphere DJF, Southern Hemisphere JJA) upper level EKE storm tracks for $1.5°C-PD$ (left) and $2.0°C-1.5°C$ (right). Top panels show responses (shading; units $m^2 s^{-2}$) along with the climatology (contour interval $40\ m^2\ s^{-2}$) for the (a) PD and (b) $1.5°C$ experiments. Bottom panels show signal-to-noise ratio $\beta/\sigma$, where the sign corresponds to the sign of the response, along with the climatology (contour interval $40\ m^2\ s^{-2}$) for the (c) $1.5°C$ and (d) $2.0°C$ experiments. In (a) and (b), hatching masks out regions where there is no model consensus (fewer than four out of five models agree) on the sign of the response. In (c) and (d), black dots mask out regions where the models do not agree on the magnitude of the response ($f^2 > 1$). The storm tracks are defined as EKE calculated from  bandpass filtered daily wind at 250 hPa.

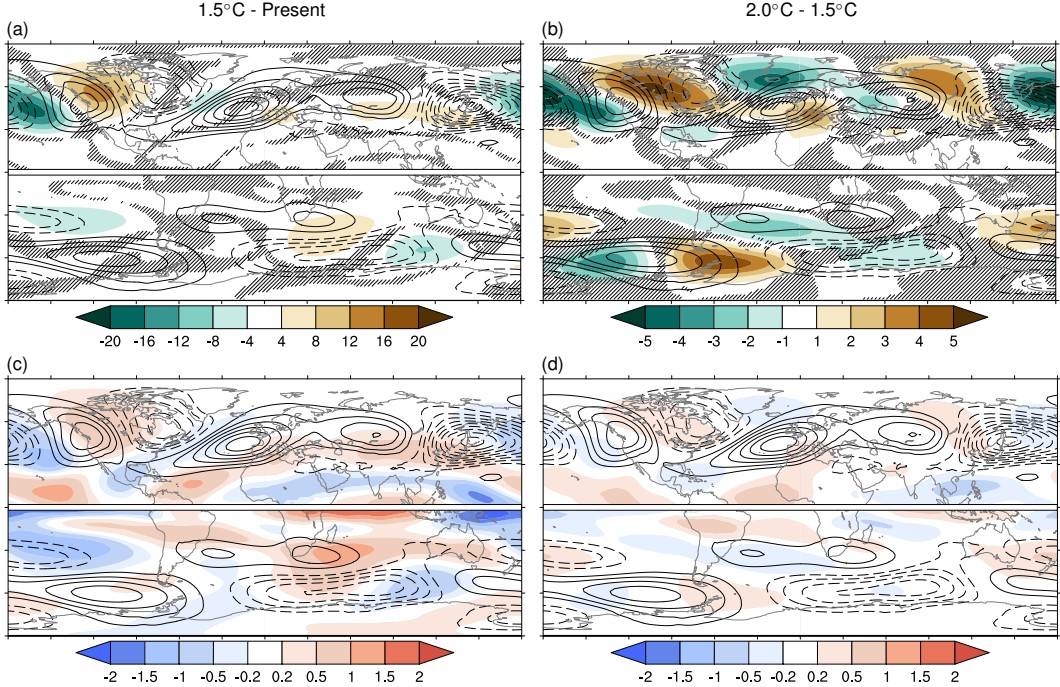

**Figure 10.** Multi-model mean response of winter (Northern Hemisphere DJF, Southern Hemisphere JJA) stationary waves at 500 hPa for $1.5°C-PD$ (left) and $2.0°C-1.5°C$ (right). Top panels show responses (shading; units m) along with the climatology (contour interval 25 m) for the (a) PD and (b) $1.5°C$ experiments. Note the different colour scale for (a) and (b). Bottom panels show signal-to-noise ratio $\beta/\sigma$, where the sign corresponds to the sign of the response, along with the climatology (contour interval 25 m) for the (c) $1.5°C$ and (d) $2.0°C$ experiments. In (a) and (b), hatching masks out regions where there is no model consensus (fewer than four out of five models agree) on the sign of the response. In (c) and (d), black dots mask out regions where the models do not agree on the magnitude of the response ($f^2 > 1$). Stationary waves are defined as departures from the zonal mean of geopotential height ($Z^*$) at 500 hPa.

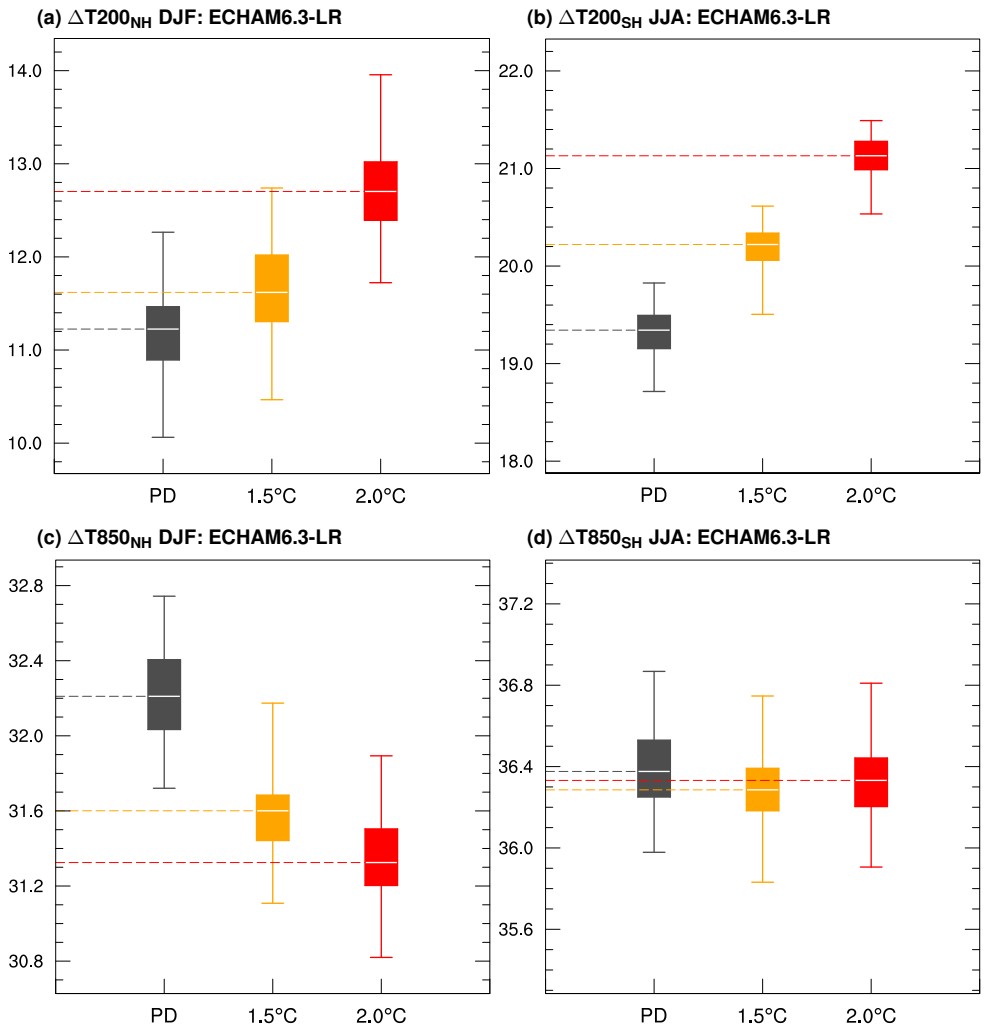

**Figure 11.** Changes in upper and lower level equator-to-pole temperature gradients (following Harvey et al., 2014) in the PD (grey), 1.5°C (orange) and 2.0°C (red) experiments in the ECHAM6.3-LR ensemble. The Northern Hemisphere temperature gradient is defined as the difference between the area-averaged temperature in the 30°S–30°N band and the region poleward of 60°N, taken at 850 hPa ($\Delta T850_{NH}$) and 200 hPa ($\Delta T200_{NH}$). The Southern Hemisphere gradient is defined as the the difference between the 30°S–30°N band and the region poleward of 60°S, taken at 850 hPa ($\Delta T850_{SH}$) and 200 hPa ($\Delta T200_{SH}$). The white horizontal line in the boxes indicates the median, the boxes indicate the interquartile range, and the whiskers indicate the ECHAM6.3-LR ensemble spread.

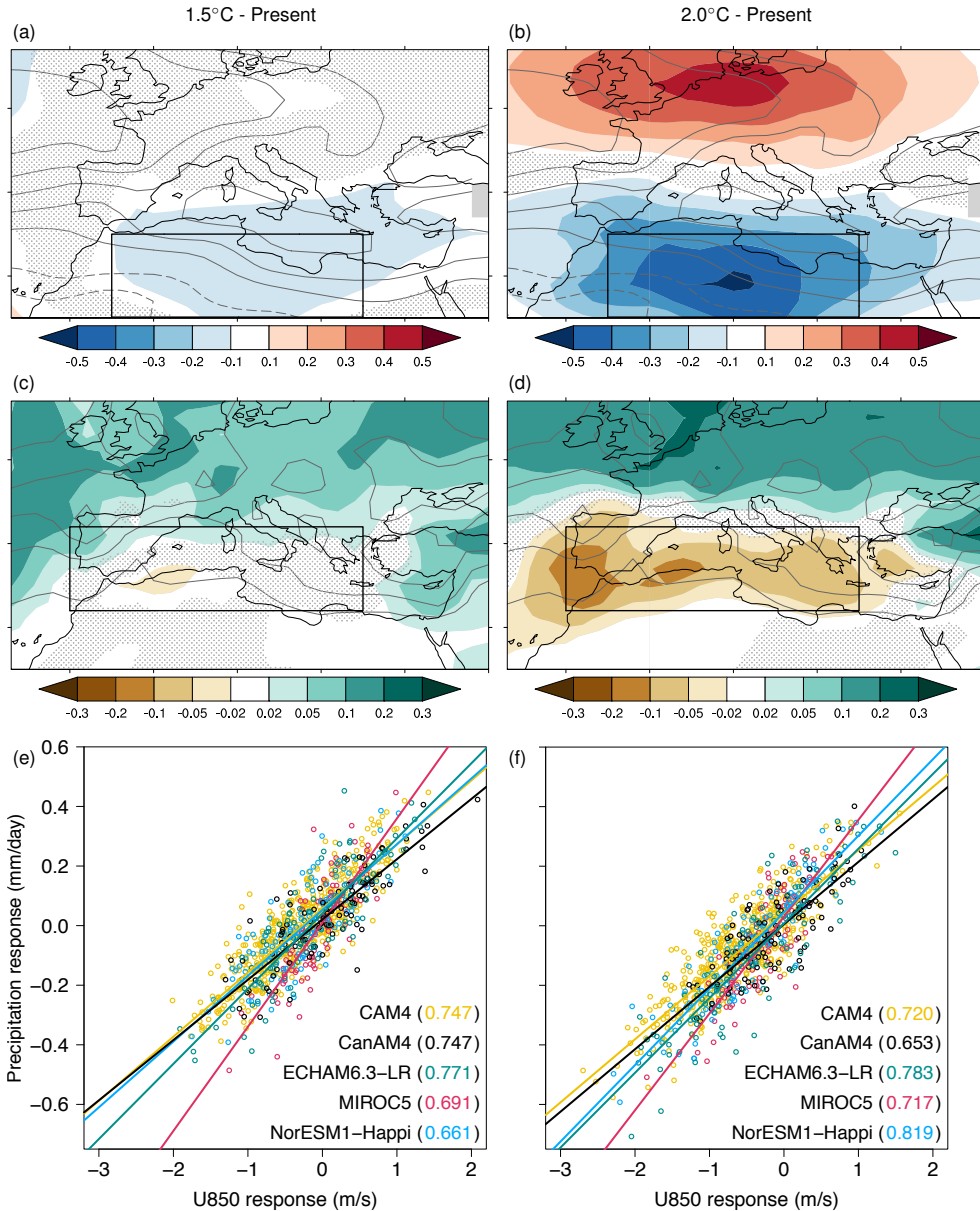

**Figure 12.** Multi-model mean winter (DJF) response of (a-b) zonal wind at 850 hPa and (c-d) precipitation in the 1.5°C (left) and 2.0°C (right) experiments relative to PD. Top panels show the multi-model mean responses (shading) and the PD climatology (contour interval 2 m s$^{-1}$, dashed contours for negative values, zero contour omitted) for $u850$. Middle panels show the multi-model mean responses (shading) and the PD climatology (contour interval 1 mm d$^{-1}$) for precipitation. Stippling in (a) to (d) masks out regions where the response is very weak (signal-to-noise ratio $|\beta/\sigma| < 0.1$). Bottom panels show the relationship between the area-averaged precipitation and $u850$ responses in the (e) 1.5°C and (f) 2.0°C experiments relative to PD, with the R$^2$ value for each model indicated in the legend. Boxes in the maps indicate the regions used for calculating area-averaged responses.

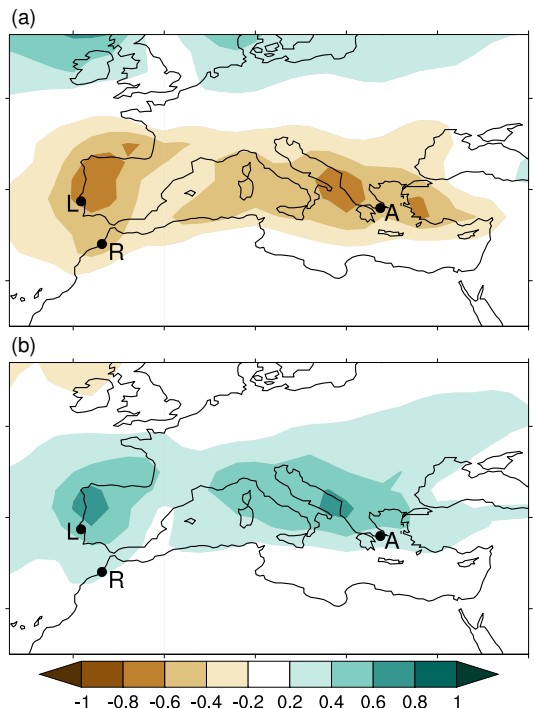

**Figure 13.** Spread in projected winter (DJF) precipitation changes for the Mediterranean. Composites of precipitation anomalies (mm d$^{-1}$) over the members with the (a) strongest 5% and (b) weakest 5% of $u850$ responses within each model ensemble in the 2.0°C experiment. Strong u850 responses are defined as those with most weakening of the mean westerlies over North Africa (Fig. 12b). Labels indicate the locations of Rabat, Lisbon and Athens.

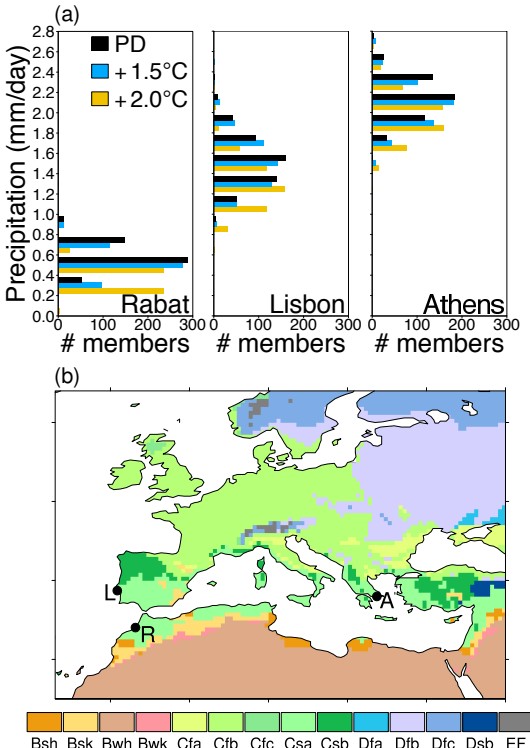

**Figure 14.** (a) Histograms of decadal mean winter (DJF) precipitation in Rabat, Lisbon and Athens for the PD, 1.5°C and 2.0°C experiments from the CAM4 ensemble. The locations are indicated in panel b as well as in Figure 13. (b) Climate zones according to the Köppen-Geiger classification scheme (Kottek et al., 2006). The Mediterranean is primarily a temperate zone with dry summers (Csa, Csb), but includes some arid regions (BSk, BSh).

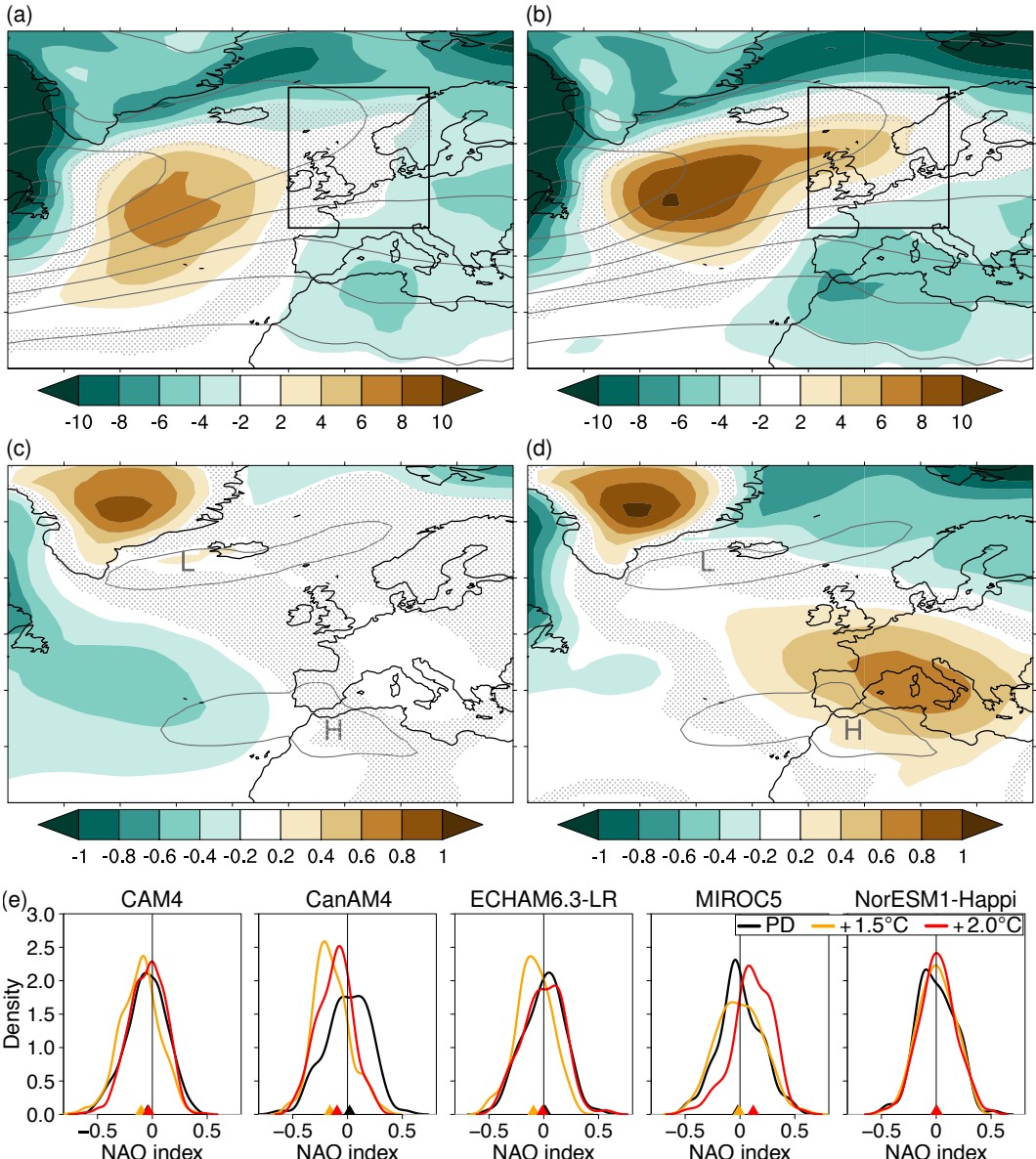

**Figure 15.** Multi-model mean response of winter (DJF) MSLP storm track and sea level pressure over the Euro-Atlantic sector in (a,c) 1.5°C and (b,d) 2.0°C relative to PD (shading, units: hPa) along with the PD climatology (contour interval 5 hPa in a,b; 1000 and 1025 hPa contours in c,d). Stippling in maps masks out regions where the response is very weak (signal-to-noise ratio $|\beta/\sigma| < 0.1$). (e) Distributions of the NAO index showing the spread and median (triangle) for each model and experiment.

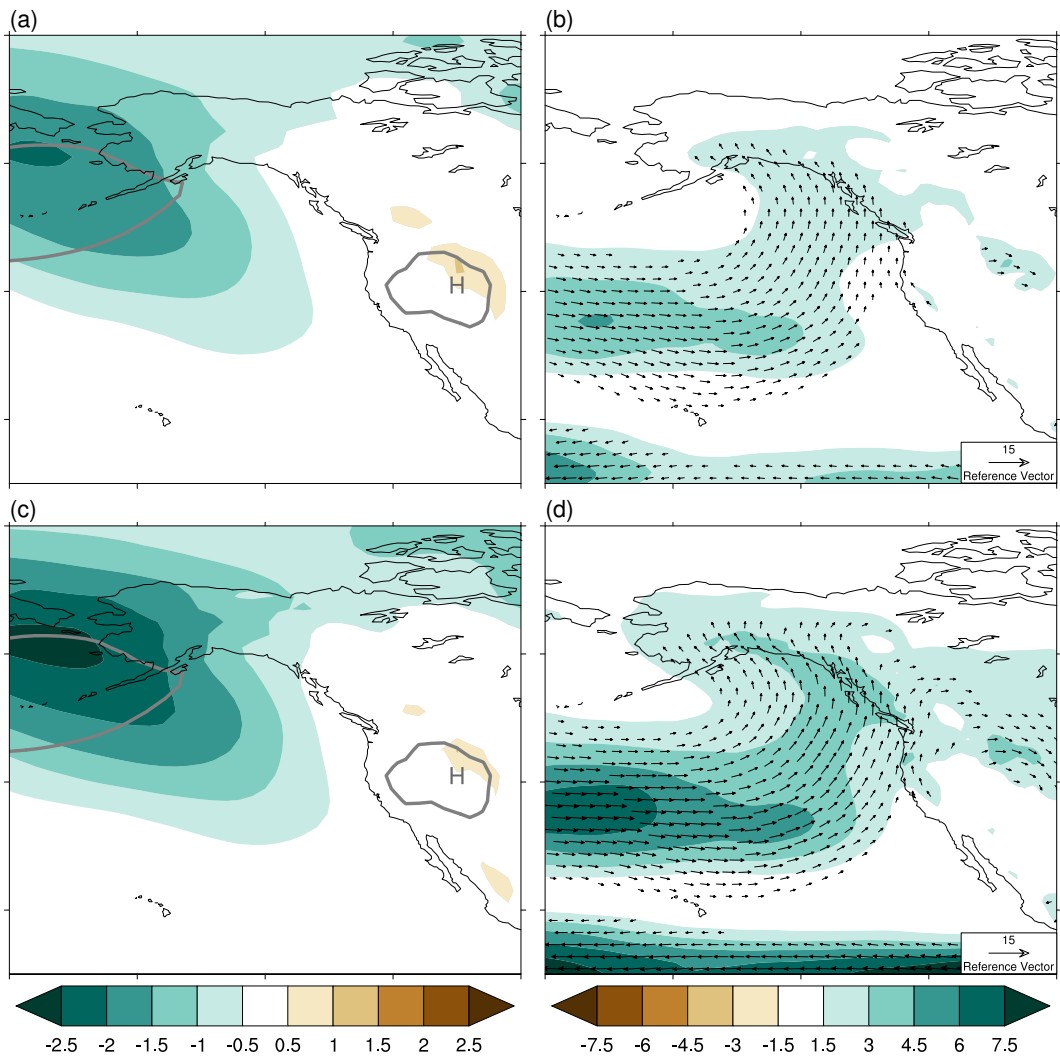

**Figure 16.** Wintertime (DJF) response of sea level pressure and moisture flux in (a,b) 1.5°C and (c,d) 2.0°C relative to PD in NorESM1-Happi. (a,c) MSLP response (shading; units hPa) along with the climatological 1000 hPa and 1025 hPa isobars from the PD experiment (grey contours). (b,d) 850-hPa specific humidity flux response (shading and vectors; units g kg$^{-1}$ m s$^{-1}$). Only vectors with magnitude greater than 2 g kg$^{-1}$ m s$^{-1}$ are plotted.

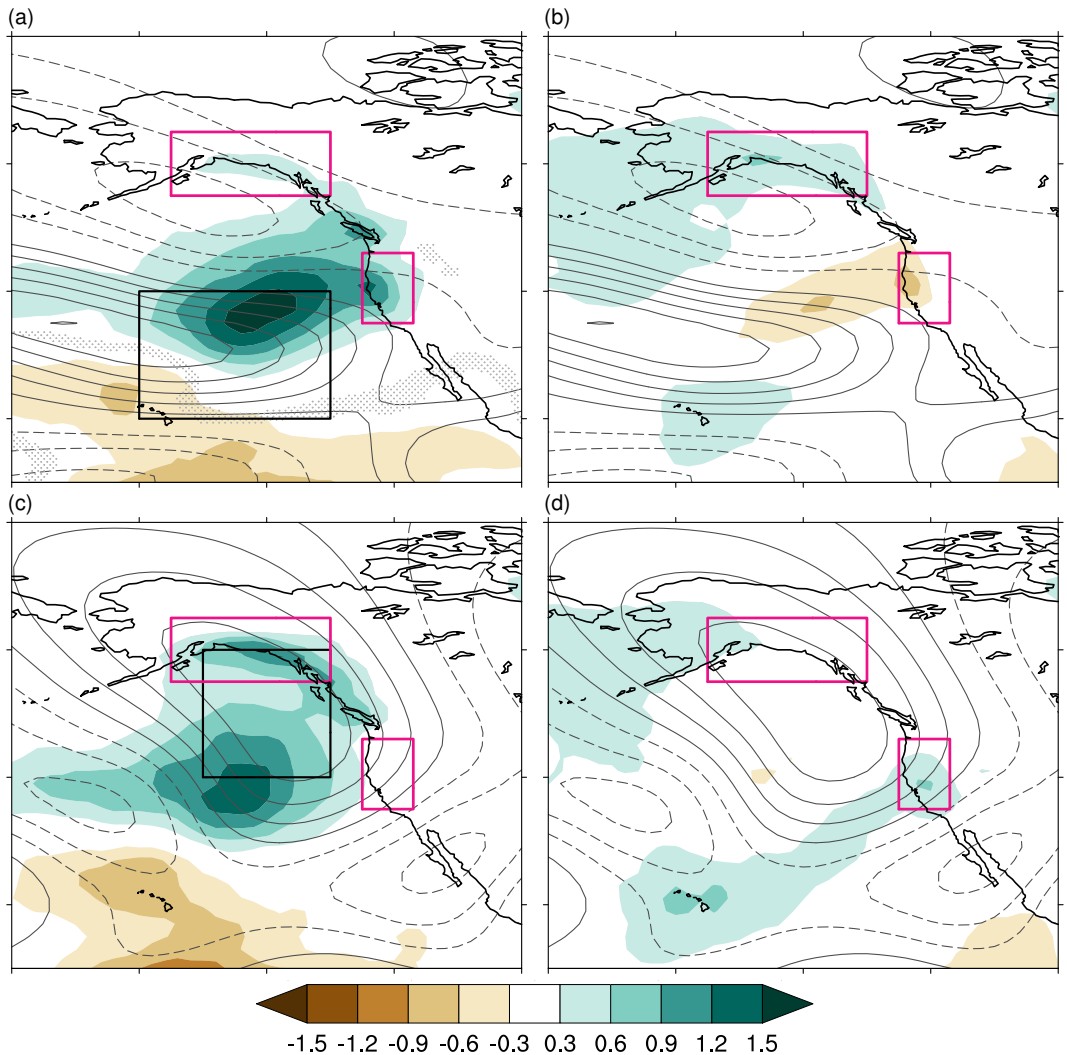

**Figure 17.** Multi-model mean precipitation signals associated with wind changes over western North America in the 2.0°C experiment during winter (DJF). Top: Composites of precipitation (units: mm day$^{-1}$) based on westerly $u250$ changes in the North Pacific jet exit (area indicated ~~in (a)~~ by ~~southern~~ the black box), over members with the (a) 5% strongest and (b) 5% weakest westerly responses. Contours show the $u250$ climatology (contour interval 0.5 m s$^{-1}$, zero contour omitted) for the PD experiment. Bottom: Composites of precipitation (units: mm day$^{-1}$) based on southerly $v250*$ changes off the coast (area indicated ~~in (a)~~ by ~~northern~~ the black box, over members with the (c) 5% strongest and (d) 5% weakest southerly responses. Contours show the $v250*$ climatology (contour interval 0.4 m s$^{-1}$, zero contour omitted) for the PD experiment, where $v250*$ is the departure from the zonal mean of the meridional wind at 250 hPa.

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
