# Peer review of "Midlatitude atmospheric circulation responses under 1.5°C and 2.0°C warming and implications for regional impacts"

_Earth System Dynamics, 2017_

## Referee Comment (RC1) · Anonymous Referee #1 · 24 Dec 2017

This study presents an analysis of the large scale circulation changes found in the HAPPI ensemble which is designed to assess the changes that occur with an additional 0.5C of warming beyond a 1.5C increase. I think this is a useful contribution and will be a beneficial resource for other users of the HAPPI ensemble. The paper is well written but I do think some aspects of the analysis and metrics could be explained more clearly as outlined in my specific comments below. The one main aspect I found confusing about the manuscript was the measure of consensus among models as outlined in my general comment and I think some improvement might be needed in this area. But overall, I think these amount to minor revisions that I recommend being made before publication.

General comments:

My main confusion lies in the metric fˆ2. This is some measure, that I recommend be explained more clearly, of the consensus among the models relative to the magnitude of the internal variability. I think this is being calculated by the ratio of the standard deviation across models to the standard deviation across members i.e., if the noise due to internal variability is bigger than the spread among ensemble means then models will be considered to agree as fˆ2 < 1. I struggle a bit to see how this is a useful metric. It seems like this could result in a situation where the model ensemble means really don't agree on even the sign of the response but the noise is sufficiently large that this metric would suggest there is consensus. I may not be fully understanding this metric as I don't think it is adequately explained. But my feeling is that models don't exhibit the degree of consensus that would be suggested by the presence of dots on the figures. As an example, if I understand correctly what's in Figure 9, this is showing the range of ensemble mean jet shifts across the models. (I'm actually not completely sure on whether it's the ensemble means or whether it's the spread acrosss all ensemble members, it's not very clear). But, if it is the ensemble mean spread, then this shows that models can range in having jet shifts of e.g., in the Pacific, -4 to +4 in DJF. Yet, the lack of dots in the North Pacific in Fig 4a tells us that there is a consensus among the models here. If the model ensemble means don't agree on the sign of the change, then I don't think defining there to be a consensus if the spread among the models is smaller than the internal variability is particularly useful. My main concern is that in almost all of the lat-lon plots, virtually all locations are described as having a strong consensus because there are no dots, but I have a hard time believing that to be the case and I think it's because this measure fˆ2 might not really be a measure of consensus, but whether the noise is bigger than the disagreement among models and in that sense I think it's misleading. I apologize if I'm misundertanding this metric, but if so, then I think it needs to be described more clearly.

Specific comments:

**[ESDD](about:blank)**
l90: About the specified SST's . Is there interannual variability or is it the climatology of that time period that is being imposed. Recommend making that clearer.

l93: It's not very clear whether the SST anomalies imposed are coming from the RCP simulations with one model e.g., the particular HAPPI model, or whether it's a CMIP5 ensemble mean. It's made clearer in the conclusions that it's coming from a CMIP5 ensemble mean but I recommend it be made clear at this point.

l131: I don't think sigma has been defined. I assume that's standard deviation, but recommend making that clear.

l139: I think f^2 should be explained in more detail rather than just referring to the Sansom paper. It's pretty unclear how this is calculated and I expect it shouldn't be too lengthy to explain. Is it just the standard deviation across models of the ensemble mean response divided by sigma?

l220: it's stated that the additional featueres in 2C compraed to 1.5C are small. But in the North Atlantic, they look pretty large. I guess it depends how you define small, but I'm not sure what the basis is for stating that the anomalies that appear in the North Atlantic are small. They're close to the magnitude of the original 1.5C anomalies over North America.

l230-232: another potentially relevant reference here is Harvey et al 2014 Equator-to-pole temperature differences and the extra-tropical storm track responses of the CMIP5 climate models. Clim Dyn, 43, 1175–1182.

l237-239: Couldn't the ensemble mean response in the CMIP5 models be pointed to for verification of this statement. It's stated that the as the world warms more the upper level temperature gradients win and we have a poleward shifting of the jet. But I don't think this is true in the east pacific during DJF where the CMIP5 models by the end of the century under RCP8.5 show a pretty good agreement on an equatorward shifting. Suggest modification of the wording to reflect this.
l263: In this discussion of the Mediterranean changes, the moisture budget analysis of Seager et al 2014 might be useful. (Seager et al 2014, Causes of Increasing Aridification of the Mediterranean Region in Response to Rising Greenhouse Gases, J. Clim., 27, 4655–4676). There the changes in P-E are decomposed into the various moisture flux contributions. Indeed the transient eddy moisture flux convergence is reduced which backs up the statements made here. But there are also substantial contributions from the altered mean flow moisture flux convergence as well.

l268: I think it would be worthwhile being more specific about where the weakening of the mean westerlies is i.e., "weakening of the mean westerlies OVER NORTH AFRICA signals" because otherwise readers might assume this is referring to weakening of westerlies over the Mediterranean which would be confusing since the Mediterranean is near the zero line of the zonal wind change. Similarly at line 279: "changes in u850 and " –> "changes in u850 over North Africa and"

l279: "of wind responses" –> "of wind responses in the 2C experiment" (because it's not clear which experiment is being referred to here).

l290: "as defined in the sense of the changes in the multi-model mean in Fig 12d" is unclear. I'm not sure exactly what this means. Does this mean that the strongest 5% are taken from all members from all models pooled together? The same goes for the caption of Fig 13.

l291: It would seem that a useful way to put this discussion of the change in the extreme percentiles into the context of a comparison with the present day climate would be to asses at what percentile does the magnitude of the 95th/5th percentile of the 2C climate occur in the PD climate. Then a statement of the form "Winters with this extreme dryness occur 5% of the time under 2C but only occur XX% of the time in PD" could be made. Otherwise, this discussion doesn't really provide any information about the change in these extremes from PD and so because of that, I don't see how it's really useful at this point. Another way to draw a comparison would be to ask how

[Figure]

much of a reduction compared to the PD climate does the dryest 5% of PD members represent i.e., a number equivalnt to the 27% that's quoted but for the driest 5% of the PD members.

Figure 1: I'm confused as to why the dots indicating a lack of conensus occur where they do. If I understand correctly, all models specify the same SST's and sea ice anomalies. If I don't understand that correctly, then I think it needs to be made clearer exactly what's done with the SSTs and sea ice. If that is correct, then I don't understand why dots are occurring around the sea ice edge and over the middle of the Pacific. I would have thought the surface air temperature would be very strongly constrained by the imposed SSTs or sea ice anoalies. If so, then why would the models differ in this region? Is it because this metric is being influence by the degree of spread among the members and there is very little spread among the members so the small spread in the response across members is actually bigger than the spread across members. This relates to my main comment above and again I wonder to what extent this metric is a useful measure of model consensus.

Figure 8 caption: it's stated that this is showing the "stationary waves". I think it would be best to be more eplicit about what is actually shown i.e., "500hPa eddy geopotential height"

Figure 9 caption: It's stated that the multi-model mean shift in the eddy driven jet for the PD is shown in grey. Firstly I don't see any grey in the figure and secondly, how would a shift be calculated for PD? I suspect this is an error in the caption and that a shift for PD isn't shown. Sorry if I'm missing it. I also think it needs to be stated more clearly whether this is the spread across ensemble means or spread across all members of all models (see my general comment above).

Figure 10 caption: I don't think this is showing "winter North Pacific eddy-driven jet" because it's showing all months of the year, not just winter.

Figure 16: suggest showing the box that's used for the composite of v in panel c rather

than panel a. I'm not sure why it makes sense to have that in panel a, but perhaps the authors have some reasoning.

Technical corrections:

l146: "show multi-model mean" –> "show the multi-model mean" l217: "weakening in southwest" –> "weakening in the southwest" l218: "strengthening in northeast" –> "strengthening in the northeast" l340: "increases over Icelend increase" –> "increases over Icelend" l371: "yield show drying" –> "yield drying" l413: suggest "investigations of how" –> "investigations into how" Figure 2 caption: "mean esponse" –> "mean response"

---

## Referee Comment (RC2) · Anonymous Referee #2 · 25 Dec 2017

General comments

This paper provides a summary of the global midlatitude circulation changes under 1.5°C and 2.0°C of warming compared to pre-industrial conditions using a multi-model ensemble of AMIP-type simulations. They focus on the winter season, and discuss various aspects of the midlatitude circulation and their influence on regional precipitation.

The experiment is well-designed with clear goals, and the authors are well-aware of limitations of the experiment.

The paper is well written and organized, and is worthy of publication after very minor revisions, although I am not quite sure if ESD is the best venue for this manuscript

considering its focus being the atmoshere, rather than interactions among earth system components as emphasized in ESD's aims and scope.

The paper is important and useful because of its rather unique focus on the near-term, limited warming scenarios. The circulation features are carefully observed and compared to numerous previous studies, which also benefit the readers. I also found the supplementary materials useful.

Minor comments/questions

L115 closing parenthsis is missing? "(Fig.1)"

L139 There are two equations that define the factor "f" in Sansom et al. (2013). I believe that you are referring to their equation (9), and the other equation (10) is not applicable to this case. Please include a specific reference to the equation to clarify your analysis method and for the readers to find relevant information more easily.

L247 I do not understand what is meant by this sentence "the influence of these anthropogenic radiative forcings is changing relative to the influence of surface warming." Could you rewrite it?

L322-323 "At low levels, the multi-model mean exhibits stronger westerlies over the continent in both warming experiments,"

Although the author states "in both warming experiments", I do not see stronger westerlies over the continent in 1.5°C warming (Fig.12a)

L334-335 It is better to mention in the main text that Fig.14c is based on one model (CanAM4). Also, does the authors see similar behaviors (stronger shift of the 95th percentile than the mean/median) in all the models? I wonder why this particular model is selected, and the results from the other models are not mentioned (unlike the previous dicussion on temperature gradient, in which the authors show an example from ECHAM6.3 in a figure and other models are summarized in tables).

L340 the word "increases" after Iceland is not needed "There is a slight increase over Iceland increases and a slight decrease over the Azores"

L371 the word "show" is not needed "the members with the weakest u250 responses yield show drying"

Fig.2 Caption r is missing in the word "response" "Multi-model mean response of winter"

Figs. 12, 14, 15, 16 Please add tick labels for latitudes and longitudes in these regional maps that have black-line contours, which make it hard to recognize geographical features drawn with gray lines. The labels may also be useful for figures 10 and 13 although they do not have black-line contours.

Figure S1 Why precipitation biases are not shown? It is one of the main variables analyzed.

---

## Author Comment (AC1) · 26 Jan 2018

Thank you to both reviewers for their careful reviews and constructive comments. We have responded to each of their points below, including additional explanations/clarifications where necessary, and a suggestion on how we would revise the manuscript to address the issue.

**Anonymous Referee #1**

This study presents an analysis of the large scale circulation changes found in the HAPPI ensemble which is designed to assess the changes that occur with an additional 0.5C of warming beyond a 1.5C increase. I think this is a useful contribution and will be a beneficial resource for other users of the HAPPI ensemble. The paper is well written but I do think some aspects of the analysis and metrics could be explained more clearly as outlined in my specific comments below. The one main aspect I found confusing about the manuscript was the measure of consensus among models as outlined in my general comment and I think some improvement might be needed in this area. But overall, I think these amount to minor revisions that I recommend being made before publication.

General comments:

My main confusion lies in the metric fˆ2. This is some measure, that I recommend be explained more clearly, of the consensus among the models relative to the magnitude of the internal variability. I think this is being calculated by the ratio of the standard deviation across models to the standard deviation across members i.e., if the noise due to internal variability is bigger than the spread among ensemble means then models will be considered to agree as fˆ2 < 1. I struggle a bit to see how this is a useful metric. It seems like this could result in a situation where the model ensemble means really don't agree on even the sign of the response but the noise is sufficiently large that this metric would suggest there is consensus. I may not be fully understanding this metric as I don't think it is adequately explained. But my feeling is that models don't exhibit the degree of consensus that would be suggested by the presence of dots on the figures. As an example, if I understand correctly what's in Figure 9, this is showing the range of ensemble mean jet shifts across the models. (I'm actually not completely sure on whether it's the ensemble means or whether it's the spread across all ensemble members, it's not very clear). But, if it is the ensemble mean spread, then this shows that models can range in having jet shifts of e.g., in the Pacific, -4 to +4 in DJF. Yet, the lack of dots in the North Pacific in Fig 4a tells us that there is a consensus among the models here. If the model ensemble means don't agree on the sign of the change, then I don't think defining there to be a consensus if the spread among the models is smaller than the internal variability is particularly useful. My main concern is that in almost all of the lat-lon plots, virtually all locations are described as having a

strong consensus because there are no dots, but I have a hard time believing that to be the case and I think it's because this measure fˆ2 might not really be a measure of consensus, but whether the noise is bigger than the disagreement among models and in that sense I think it's misleading. I apologize if I'm misunderstanding this metric, but if so, then I think it needs to be described more clearly.

The f^2 metric comes from the Sansom et al. 2013 ANOVA (analysis of variance) framework. We explain more about its derivation and interpretation below. In summary: 1) f^2 is a bit complicated, so we understand the reviewer's confusion given our terse explanation, 2) we suggest adding some text to the methods as well as about its interpretation, and including an appendix with an abbreviated derivation of the metric within the ANOVA framework, 3) we still feel it provides some useful information so would like to keep it, and 4) we are happy to add a traditional measure of consensus to the figures as well.

The f^2 metric comes from a statistical test for "the presence of model dependence in the climate response" (Sansom et al. 2013, equation 9). It comes from the ANOVA framework, which allows one to break down a multi-model ensemble response to distinguish between the expected climate change by the entire ensemble in a certain scenario, and the "structural" or model dependence. Specifically, f^2 is the ratio of the variance explained by structural uncertainty (model dependence) in the climate response to that explained by internal variability, but it is not simply the ratio of standard deviation across models to the standard deviation across members. It is defined as $f^2 = (R\_gamma^2 - R\_alpha^2)/(1-R\_gamma^2)$. The $R^2$ variables are "coefficients of determination", representing the proportion of total variability explained by a normal linear regression framework in two situations: $R\_alpha$ is for the case assuming no model dependence of the response, and $R\_gamma$ is for the case assuming there is model dependence. The $(R\_gamma^2 - R\_alpha^2)$ in the numerator of $f^2$ is the variability due to model dependence, in other words, the variation (sum of squares) in the fitted values. The $(1-R\_gamma^2)$ in the denominator is simply the variability from internal variability (for HAPPI, this would be the 10-year mean from each individual ensemble).

f^2 tells us how large the uncertainty in the forced response is compared to the variation in the 10-year mean climate; it does not take into account the sign of the response. In other words, it provides information on how well the models agree on the *magnitude* of the climate change response compared to internal variability. For this reason, we state (l141-143): "We interpret $f^2$ < 1 as evidence of consensus on the magnitude of the climate change response, as this implies that internal climate variability is the dominant source of uncertainty in the multi-model projections." This metric has previously been used as a measure of consensus in multi-model projections by Zappa et al. 2013. Perhaps the wording is too subtle and/or too easily misconstrued; we could just avoid using the word "consensus" when referring to $f^2$.

f^2 provides information that is complementary to the standard sign agreement among models . As stated in Zappa et al. 2013, sign agreement "[tends] to systematically reject consensus where the mean climate response is small relative to the internal variability (i.e., in regions of low signal-to-noise ratio). However, additional information can be gained from a statistical

analysis in regions of small mean climate response if climate models agree that the response is small." If model dependence is proportionally large ($f^2>1$), members with the same sign of response may not be considered to agree on the magnitude of the response; If model dependence is proportionally small ($f^2<1$), members with different signed responses may still agree that there is weak (or no) forced response. The issue is also discussed in detail in the IPCC AR5 Chapter 12, Box 12.1, which describes different methods for quantifying model consensus, and what information these methods provide (or don't provide).

For the HAPPI ensemble, there should be less of an issue with no consensus where the mean climate response is small because there are so many members (the forced response is very well sampled). Some considerations are that a) $f^2$ is sensitive to the length of the time slice (longer time slices give smaller internal variability, which inflates $f^2$), and b) the HAPPI ensemble has no decadal SST variability, which would also tend to inflate $f^2$ (this is probably mostly important in the tropics rather than the extratropics).

We suggest adding a standard measure of consensus to the upper panels of Fig. 2-8 to indicate consensus (or lack of consensus) on the direction of change and leave the $f^2$ metric in the lower panels (see example below). We would better explain the $f^2$ metric and its caveats in the methodology section, and be more careful with its interpretation where applicable.

See the response to the comment below about Figure 9 caption for more on the apparent discrepancy between Figure 9 and Figure 4a.

Sample figure equivalent to left panels of Fig. 5 for U250 in 1.5-PD: (a) U250 response (shading), PD climatology (contours), and model consensus (at least 4/5 models agree on the sign of the forced response where there is no stippling). (b) signal-to-noise ratio beta/sigma (shading) and $f^2>1$ (dots). There are no dots here, which means that internal variability is the main source of uncertainty in the climate change response everywhere for this field. Over Eurasia, it is white with no dots in panel (b), which means that the models agree the forced response is small, even though there is no consensus on the sign of the change (stippling in panel (a)). (Note that there are slight differences in the amplitude of the response compared to the figure in the submitted manuscript because it was discovered after submission that some members of one of the models contained a bug. These members have been rerun/corrected in the figure below.)

(a)

[Figure]

-2.5  -2  -1.5  -1  -0.5  0.5  1  1.5  2  2.5

(b)

[Figure]

-2  -1.5  -1  -0.5  -0.2  0.2  0.5  1  1.5  2

Specific comments:

l90: About the specified SST's . Is there interannual variability or is it the climatology of that time period that is being imposed. Recommend making that clearer.

There is interannual variability in the the specified SSTs.  We will clarify this.

l93: It's not very clear whether the SST anomalies imposed are coming from the RCP simulations with one model e.g., the particular HAPPI model, or whether it's a CMIP5 ensemble mean. It's made clearer in the conclusions that it's coming from a CMIP5 ensemble mean but I recommend it be made clear at this point.

We can add some more detail about how the 1.5C and 2C experiment SST boundary conditions were constructed, rather than just referring to the Mitchell et al. methodology paper. This should help clarify the issue. It's true, as the reviewer gleaned from the conclusions, that they forcing comes from the CMIP5 ensemble mean, so the models are forced with identical boundary conditions.

l131: I don't think sigma has been defined. I assume that's standard deviation, but recommend making that clear.

Sorry, we did indeed forget to define sigma. It's the standard deviation of the means of the individual members. Since each member is one decade long, this gives a measure of the noise from decadal variability. We will clarify this in the text.

l139: I think f^2 should be explained in more detail rather than just referring to the Sansom paper. It's pretty unclear how this is calculated and I expect it shouldn't be too lengthy to explain. Is it just the standard deviation across models of the ensemble mean response divided by sigma?

We will add some more text to describe the metric and its interpretation. It's not too lengthy to explain, but is a little different than one might expect because the variability estimates come from the ANOVA framework (see response to General comments above).

l220: it's stated that the additional features in 2C compared to 1.5C are small. But in the North Atlantic, they look pretty large. I guess it depends how you define small, but I'm not sure what the basis is for stating that the anomalies that appear in the North Atlantic are small. They're close to the magnitude of the original 1.5C anomalies over North America.

This is a good point. The features are probably overemphasized because of the map projection, but they are fairly large amplitude, as the reviewer points out. We will remove the "rather small" qualifier.

l230-232: another potentially relevant reference here is Harvey et al 2014 Equator-to- pole temperature differences and the extra-tropical storm track responses of the CMIP5 climate models. Clim Dyn, 43, 1175–1182.

We have used this reference in other places in the manuscript, but it is certainly very relevant here as well. We will add it.

l237-239: Couldn't the ensemble mean response in the CMIP5 models be pointed to for verification of this statement. It's stated that the as the world warms more the upper level temperature gradients win and we have a poleward shifting of the jet. But I don't think this is true in the east pacific during DJF where the CMIP5 models by the end of the century under RCP8.5 show a pretty good agreement on an equatorward shifting. Suggest modification of the wording to reflect this.

The CMIP5 models should indeed be used here. We can add a reference to Barnes and Polvani 2013 Fig. 12, which shows a poleward shift of the zonal-mean jets in the North Atlantic, North Pacific and Southern Hemisphere in RCP8.5. It's true that in that the winter North Pacific jet exit (eastern North Pacific) actually shows an equatorward shift (Simpson et al. 2012, Fig. 8). The text will be edited to include this information.

l263: In this discussion of the Mediterranean changes, the moisture budget analysis of Seager et al 2014 might be useful. (Seager et al 2014, Causes of Increasing Aridification of the Mediterranean Region in Response to Rising Greenhouse Gases, J. Clim., 27, 4655–4676). There the changes in P-E are decomposed into the various moisture flux contributions. Indeed the transient eddy moisture flux convergence is reduced which backs up the statements made here. But there are also substantial contributions from the altered mean flow moisture flux convergence as well.

Thank you for alerting us to this relevant study containing a breakdown of the Mediterranean moisture budget. We will include a discussion of the mean flow moisture convergence as an important contribution to Mediterranean drying.  The period of interest in Seager et al. is 2021-2040 of RCP8.5, which actually corresponds quite well to the HAPPI low warming scenarios. The mean circulation changes (Figures 9 and 10 of the paper) are consistent with the HAPPI 2C experiment, including an extension of the storm track into Europe and an anomalous near-surface high over the central Mediterranean (Z850 field in Seager et al. Fig. 10, MSLP in our manuscript Fig. 15b) that is linked to subsidence and mass flux divergence.

l268: I think it would be worthwhile being more specific about where the weakening of the mean westerlies is i.e., "weakening of the mean westerlies OVER NORTH AFRICA signals" because otherwise readers might assume this is referring to weakening of westerlies over the Mediterranean which would be confusing since the Mediterranean is near the zero line of the zonal wind change. Similarly at line 279: "changes in u850 and " –> "changes in u850 over North Africa and"

We agree, and will add specific geographical indicators as suggested.

l279: "of wind responses" –> "of wind responses in the 2C experiment" (because it's not clear which experiment is being referred to here).

We agree, this is clearer and we will edit accordingly.

l290: "as defined in the sense of the changes in the multi-model mean in Fig 12d" is unclear. I'm not sure exactly what this means. Does this mean that the strongest 5% are taken from all members from all models pooled together? The same goes for the caption of Fig 13.

We were worried that it might be confusing to define a "strong" response as one where the winds become weakest over the Mediterranean box. This extra phrase was meant to clarify, but perhaps it would be better to leave it out. (We have included an additional figure in the response to the next comment that may help to clarify as well.) All members from all models are considered for the strongest and weakest 5% composites in Figure 13, but the extremes are identified in each model first, then composited. There is not too much difference if all members from all models are pooled - this can be seen already from the scatterplots in Fig. 12 e/f, where the models generally cover the range from weak to strong U850 responses (CanAM4 sits a bit towards the "strong response" end, and MIROC5 sits a bit towards the "weak response" end).

l291: It would seem that a useful way to put this discussion of the change in the extreme percentiles into the context of a comparison with the present day climate would be to asses at what percentile does the magnitude of the 95th/5th percentile of the 2C climate occur in the PD climate. Then a statement of the form "Winters with this extreme dryness occur 5% of the time under 2C but only occur XX% of the time in PD" could be made. Otherwise, this discussion doesn't really provide any information about the change in these extremes from PD and so because of that, I don't see how it's really useful at this point. Another way to draw a comparison would be to ask how much of a reduction compared to the PD climate does the dryest 5% of PD members represent i.e., a number equivalent to the 27% that's quoted but for the driest 5% of the PD members.

Thanks for these very nice suggestions. We have done some preliminary analyses along these lines (see below). We can refine these analyses and include some subset of them in the revised manuscript in order to highlight the change in the extreme percentiles of the precipitation distribution. Before continuing to this, we'd just like to clarify a possible misunderstanding. The statement about the 27% reduction and Figure 13 do in fact refer to extremes, but extremes in the distribution of 2C-PD wind response rather than in the distribution of the 2C and PD precipitation. Specifically, the 27% reduction comes from the area-averaged precipitation anomalies in the "strong wind response" (Fig. 13a) compared to the climatological precip in the PD experiment. In case this isn't clear, below, we show a version of Fig. 12f for one model only, with the members colour-coded to help illustrate this. The "strong wind response" members are those for which the westerlies weaken most, i.e., everything to the left of the vertical dotted line (blue + yellow members in bottom left quadrant). The "27% reduction" in the manuscript is the anomalous drying of these members (-0.372 mm/day) as a percentage of the mean regional precipitation in the PD experiment (1.349 mm/day). This was meant to give some feeling for how large a percentage change in precip can be in members with the most drying.

This is indeed different from the information the reviewer is asking for, but we wanted to clarify this as it is relevant to our response to the reviewer's second suggestion below.

[Figure]

Regarding the reviewer's first suggestion, the table below documents how the extreme percentiles of area-averaged precipitation over the Mediterranean region in the 2C experiment map onto the PD distribution - that is, the PD percentiles that correspond to the 5th and 95th percentiles of the 2C experiment. Taking a member-weighted mean across the models, we could make this statement: *Extremely dry winters corresponding to the 5th precipitation percentile in the 2C experiment occur <1% of the time in PD on average* (with considerable model spread - note that CanAM4 does not show any change in this respect). (Also: Extremely wet winters corresponding to the 95th precipitation percentile in the 2C experiment are nearly three times as likely (14.5%) in the PD on average.) An additional refinement could be to split out the northern region of the box, which actually experiences wetting in the 1.5C experiment, and calculate changes based only on the areas that dry.

| | CAM4-2degree | CanAM4 | ECHAM6-3-LR | MIROC5 | NorESM1-HAPPI | Member-weighted mean |
|---|---|---|---|---|---|---|
| 5th percentile (5% driest) | 0% | 5% | 1% | 0% | 0.8% | 0.76% |
| 95th percentile (5% wettest) | 83.4% (16.6%) | 89% (11%) | 77% (23%) | 92% (8%) | 94.4% (5.6%) | 85.5% (14.5%) |

Regarding the reviewer's second suggestion: As an example calculation for the model shown in the scatterplot above, the "dry extremes" would show an 11% reduction in precipitation from the PD to the 2C experiment, while the "wet extremes" show only a 3% reduction. As we explain above, this isn't really equivalent to the 27% quoted in the text, as the 27% relates to extremes of the wind response distribution. However, this could indeed be a good point to make in the text, as the drying in the 5th percentile extremes is nearly twice as much as the mean drying in the region. We can add these results to the text.

For interest, the precipitation amounts (mm/day) associated with all the "wind response", "precip response" and straight precipitation distributions we have been discussing for the the scatterplot above are:

Mean/Median Mediterranean precip in 2C experiment:
Mean: 1.265                    Median: 1.266
Mean/Median Mediterranean precip in PD experiment:
Mean: 1.349                    Median: 1.353
Mean precip of 5% and 95% extremes based on wind response (blue+yellow dots) in 2C-PD:
5%: -0.372                    95%: 0.200
Mean precip of 5% and 95% extremes based on precip response (blue+red dots) in 2C-PD:
5%: -0.413                    95%: 0.241
Mean precip of 5% and 95% precip extremes in 2C experiment:
5%: 1.025                    95%: 1.508
Mean precip of 5% and 95% precip extremes in PD experiment:
5%: 1.151                    95%: 1.556

Figure 1: I'm confused as to why the dots indicating a lack of consensus occur where they do. If I understand correctly, all models specify the same SST's and sea ice anomalies. If I don't understand that correctly, then I think it needs to be made clearer exactly what's done with the SSTs and sea ice. If that is correct, then I don't understand why dots are occurring around the sea ice edge and over the middle of the Pacific. I would have thought the surface air temperature would be very strongly constrained by the imposed SSTs or sea ice anomalies. If so, then why would the models differ in this region? Is it because this metric is being influence

by the degree of spread among the members and there is very little spread among the members so the small spread in the response across members is actually bigger than the spread across members. This relates to my main comment above and again I wonder to what extent this metric is a useful measure of model consensus.

This confusion stems from the misunderstanding about what the f^2 metric means, and we believe that most of it is covered in our response to the General comments above. Ocean points are indeed more strongly constrained by the specified SST/sea ice conditions, so internal variability will be smaller here, and hence may not dominate the multi-model mean response in the ANOVA framework. If we add a standard measure of consensus (sign agreement) to the top panels as suggested above, this should help the interpretation. All models do indeed specify the same SSTs and sea ice.

Figure 8 caption: it's stated that this is showing the "stationary waves". I think it would be best to be more explicit about what is actually shown i.e., "500hPa eddy geopotential height"

We will specify "Stationary waves are defined as departures from the zonal mean of geopotential height (Z*) at 500 hPa." We'd like to keep the term "stationary waves" in the caption as well, as this may be more familiar to non-specialist readers.

Figure 9 caption: It's stated that the multi-model mean shift in the eddy driven jet for the PD is shown in grey. Firstly I don't see any grey in the figure and secondly, how would a shift be calculated for PD? I suspect this is an error in the caption and that a shift for PD isn't shown. Sorry if I'm missing it. I also think it needs to be stated more clearly whether this is the spread across ensemble means or spread across all members of all models (see my general comment above).   [From General comments above: As an example, if I understand correctly what's in Figure 9, this is showing the range of ensemble mean jet shifts across the models. (I'm actually not completely sure on whether it's the ensemble means or whether it's the spread across all ensemble members, it's not very clear). But, if it is the ensemble mean spread, then this shows that models can range in having jet shifts of e.g., in the Pacific, -4 to +4 in DJF. Yet, the lack of dots in the North Pacific in Fig 4a tells us that there is a consensus among the models here. If the model ensemble means don't agree on the sign of the change, then I don't think defining there to be a consensus if the spread among the models is smaller than the internal variability is particularly useful.]

We apologize, the caption is incorrect - these are indeed the shifts relative to PD. This is the spread across the entire HAPPI ensemble (all members, all models), not the spread across ensemble means for each model. Taking the North Pacific DJF jet shift as mentioned by the reviewer in the General comments, we see a large spread, with a slightly negative (equatorward shift) ensemble mean. In Fig. 10, we see mostly blue across the North Pacific in DJF indicating a slight equatorward shift in the ensemble mean, consistent with Fig. 9b.  The lack of dots in the North Pacific in Fig. 4a is because the models agree that the uncertainty in the forced response is small compared to internal variability (variations in the 10-year means) - see response to

General comments above. The small signal-to-noise in the region reflects the large spread across all ensemble members shown in Fig. 9b.

The caption will be clarified/edited accordingly and some additional discussion added to make the link between results presented in various figures where needed.

Figure 10 caption: I don't think this is showing "winter North Pacific eddy-driven jet" because it's showing all months of the year, not just winter.

The caption is incorrect here as well. We will correct it.

Figure 16: suggest showing the box that's used for the composite of v in panel c rather than panel a. I'm not sure why it makes sense to have that in panel a, but perhaps the authors have some reasoning.

Yes, we agree. The boxes were probably all left in panel a as we were developing and changing the figure. We will move it to panel c.

Technical corrections:

l146: "show multi-model mean" –> "show the multi-model mean" l217: "weakening in southwest" –> "weakening in the southwest" l218: "strengthening in northeast" –> "strengthening in the northeast" l340: "increases over Icelend increase" –> "increases over Iceland" l371: "yield show drying" –> "yield drying" l413: suggest "investigations of how" –> "investigations into how" Figure 2 caption: "mean esponse" –> "mean response"

Thank you for catching these errors. They will all be corrected.

**Anonymous Referee #2**

General comments

This paper provides a summary of the global midlatitude circulation changes under 1.5◦C and 2.0◦C of warming compared to pre-industrial conditions using a multi-model ensemble of AMIP-type simulations. They focus on the winter season, and discuss various aspects of the midlatitude circulation and their influence on regional precipitation.

The experiment is well-designed with clear goals, and the authors are well-aware of limitations of the experiment.

The paper is well written and organized, and is worthy of publication after very minor revisions, although I am not quite sure if ESD is the best venue for this manuscript considering its focus being the atmosphere, rather than interactions among earth system components as emphasized in ESD's aims and scope.

The paper is important and useful because of its rather unique focus on the near- term, limited warming scenarios. The circulation features are carefully observed and compared to numerous previous studies, which also benefit the readers. I also found the supplementary materials useful.

The focus of this manuscript is indeed the atmosphere, but we feel there are good reasons to include it in this special issue of ESD on "The Earth system at a global warmin of 1.5C and 2.0C" ( https://www.earth-syst-dynam.net/special_issue909.html ). The HAPPI initiative addresses "earth system" implications of the 1.5C versus 2C warming targets, including impacts related to management of the earth system and interactions with the biosphere (subject areas 3 and 4 of ESD: https://www.earth-system-dynamics.net/about/aims_and_scope.html) as well as scenarios of future climatic change and climate predictions (subject area 2). This manuscript describes the large-scale circulation changes that underpin more impacts-focused HAPPI studies, many of which also appear in this special issue. In addition, some of the other studies in this special issue are rather atmosphere-focused, including Wehner et al. (accepted) on tropical cyclones, Barcikowska et al. (in review) on Euro-Atlantic storminess .

Minor comments/questions

L115 closing parenthesis is missing? "(Fig.1)"

We will correct this.

L139 There are two equations that define the factor "f" in Sansom et al. (2013). I believe that you are referring to their equation (9), and the other equation (10) is not applicable to this case. Please include a specific reference to the equation to clarify your analysis method and for the readers to find relevant information more easily.

We are using the f defined in equation 9, and also used for example in Zappa et al. 2013 "A Multimodel Assessment of Future Projections of North Atlantic and European Extratropical Cyclones in the CMIP5 Climate Models". More details on the issue of model consensus will be added to the text (see also response to Reviewer 1's General comments above).

L247 I do not understand what is meant by this sentence "the influence of these anthropogenic radiative forcings is changing relative to the influence of surface warming." Could you rewrite it?

This should probably be expanded upon to make it clearer.  In the 2.0C experiment, $CO_2$ concentrations are higher than those in the 1.5C experiment, and the SST boundary conditions are warmer than those in the 1.5C experiment. However, the "other" forcings listed in the first

part of the sentence (aerosols, ozone, land use) are the same in the two warming experiments. So the influence of the "other forcings" relative to, say, CO2 is different in 1.5C versus 2.0C, but we shouldn't have said that it was relative to "surface warming". For example, the warming of the Antarctic stratosphere due to ozone recovery is the same in the two experiments, while the radiative effects of CO2 will be stronger in 2.0C than in 1.5C. A suggested edit: "In the 2.0C experiment, atmospheric CO2 concentrations are higher and SSTs are warmer than in the 1.5C experiment; however, aerosols, ozone and land use are set to the same values in the two warming experiments (taken from the year 2095 in the RCP2.6 scenario). Thus, the influence of CO2 and surface warming is changing relative to the influence of other anthropogenic radiative forcings."

L322-323 "At low levels, the multi-model mean exhibits stronger westerlies over the continent in both warming experiments,"  Although the author states "in both warming experiments", I do not see stronger westerlies over the continent in 1.5°C warming (Fig.12a)

Our apologies, we seem to have mixed up the two statements. The weaker westerlies over the Mediterranean are present in both experiments, while the stronger westerlies over Europe are only in the 2.0C experiments. We will correct this.

L334-335 It is better to mention in the main text that Fig.14c is based on one model (CanAM4). Also, does the authors see similar behaviors (stronger shift of the 95th percentile than the mean/median) in all the models? I wonder why this particular model is selected, and the results from the other models are not mentioned (unlike the previous discussion on temperature gradient, in which the authors show an example from ECHAM6.3 in a figure and other models are summarized in tables).

We agree that it would be more useful to present the results for all the models rather than just one.  It is somewhat challenging to define consistent metrics because the precipitation patterns, including the region where Mediterranean drying appears in the 2.0C experiment, vary somewhat from model to model, and this is especially true for the "extremes" (defined here as the 95th percentile precipitation rate). However, it generally holds that the change in mean precipitation is comparable in 1.5C-PD and 2C-1.5C (top row shows multi-model mean precipitation change).  For the 95th percentile precipitation, the multi-model mean shows slightly larger areas of substantial responses, even given the intermodel spread in the patterns and the fact that the original box we defined is perhaps not ideal (bottom row).

We suggest removing the model-specific panel in Fig. 14c and replacing it with a figure that shows results from all the models. For this, we would refine the analysis to address some of the issues above - for example, we could do an area average over the grid points where the MMM shows an increase in precipitation only, or over land only, and/or we could adjust the region based on the MMM patterns. The MMM could be shown in the figure, and results for individual models could be tabulated if they cannot be shown simply and clearly in the figure.

Top row: Multi-model mean response in winter averaged precipitation (mm/day) for 1.5-PD (left) and 2-1.5 (right). Bottom row: Multi-model mean response in 95th percentile precipitation (mm/day, same colour scale as top row).

[Figure]

L340 the word "increases" after Iceland is not needed "There is a slight increase over Iceland increases and a slight decrease over the Azores"

We will correct this.

L371 the word "show" is not needed "the members with the weakest u250 responses yield show drying"

We will correct this.

Fig.2 Caption r is missing in the word "response" "Multi-model mean response of winter"

We will correct this.

Figs. 12, 14, 15, 16 Please add tick labels for latitudes and longitudes in these regional maps that have black-line contours, which make it hard to recognize geographical features drawn with gray lines. The labels may also be useful for figures 10 and 13 although they do not have black-line contours.

Yes, we agree that the continent outlines do not show up well on many of these figures. We will play around with labels and the plots themselves to improve this. We can also add a note about the region being shown in the caption.

Figure S1 Why precipitation biases are not shown? It is one of the main variables analyzed.

This is a good point. We will add the precipitation biases to the Supplement.